# Informal sector employment and the health outcomes of older workers in India

**Poulomi Chowdhury[1]\*, Itismita Mohanty[1], Akansha Singh[2], Theo Niyonsenga[1]**

**1** Health Research Institute, Faculty of Health, University of Canberra, Canberra, Australian Capital Territory, Australia, **2** Department of Anthropology, Durham Research Methods Centre, Durham University, Durham, United Kingdom

\* poulomi.chowdhury@canberra.edu.au

**Data Availability Statement:** The data underlying this study are third party and were collected by the Longitudinal Ageing Study in India (2017-18). Raw survey data have been accessed from International

## Abstract

A large proportion of the older population in India constitutes an undeniable share of work-force after the retirement age. This stresses the need to understand the implications of working at older ages on health outcomes. The main objective of this study is to examine the variations in health outcomes by formal/informal sector of employment of older workers using the first wave of the Longitudinal Ageing Study in India. Using binary logistic regression models, the results of this study affirm that type of work does play a significant role in determining health outcomes even after controlling socio-economic, demographic, life-style behaviour, childhood health and work characteristics. The risk of Poor Cognitive Functioning (PCF) is high among informal workers, while formal workers suffer greatly from Chronic Health Conditions (CHC) and Functional Limitations (FL). Moreover, the risk of PCF and/or FL among formal workers increases with the increase in risk of CHC. Therefore, the present research study underscores the relevance of policies focusing on providing health and healthcare benefits by respective economic activity and socio-economic position of older workers.

## Introduction

Globally, ageing population has become a common phenomenon owing to reduction in fertility and mortality rates, and these integrated effects have altered the age-sex composition as well as labour force participation towards higher ages [1, 2]. Speculation indicates that the world population aged 60+ will be around 2 billion in 2050 from 900 million in 2015 [3]. By 2050, nearly 80 percent of the older population will reside in low-middle income countries [3]. Ageing process in developed nations has slowed down the economic growth through insufficient labour force. These nations have implemented policies aiming to boost labour force market by motivating the older population to work past the age of 65 years [4–10]. Consequently, in recent years, the share of older workers has amplified substantially in developed nations [4, 5, 7, 9–13].

Apart from these nations, the developing countries are also experiencing shifts in age-structures with a tremendous pace. United Nations (UN) report (2017), states that almost 60 percent of the older people are currently living in developing countries, and is growing rapidly

Institute for Population Sciences website, and the data request form has been downloaded from the following link: https://www.iipsindia.ac.in/content/LASI-data. Then, the duly signed form has been sent to the email "datacenter@iipsindia.ac.in" for approval. The principal coordinator of LASI data is Prof. T. V. Sekher (email: tvsekher@iipsindia.ac.in). The authors confirm that others would be able to access the data in the same manner and that the authors did not have any special access privileges that others would not have.

**Funding:** The author(s) received no specific funding for this work.

**Competing interests:** The authors have declared that no competing interests exist.

compared to developed countries [14]. Alas, the provision and implementation of pension benefits or retirement programs are less prevailing in developing nations. Apparently, only 20 percent of this population is entitled to any pension related benefits, yet most of them rely on family support system [15]. Accordingly, significant proportion of older people are active in labour market in developing countries [2, 16–20] than developed countries [21].

India, the second most populous nation in the world, is also facing remarkable increase in ageing population, reportedly 8.6 percent (104 million) of the total population in 2011 [22]. This figure is expected to escalate to about 20 percent and, in terms of absolute numbers, the country will be a home for 319 million older individuals by 2050 [23]. Soon this older population will surpass the young population below 14 years [24].

Ageing is normally associated with chronic health conditions which upsurge later in life [22, 25–27]. As evident from prior studies, more than half of the older people endure non-communicable diseases (NCDs), while one-fourth are affected by multimorbidity [28, 29]. Across India, these chronic diseases exhibit huge heterogeneity in terms of socio-economic conditions, place of residence and gender [22, 26, 30]. Further, estimated figures suggest that the burden of NCDs will constitute a large share of the national disability [31]. Projected numbers demonstrate that roughly 45 percent of the health burden will be borne by older people [32, 33], making the health requirements of older people comparatively higher than other age groups.

Specific social security, poverty alleviation and social welfare programs have been launched to address the challenges associated with older population's social and health conditions. Certain policies and programs implemented by the ministries, namely, National Policy on Older Persons, National Social Assistance Programme, National Policy for Senior Citizens, Indira Gandhi National Old Age Pension Scheme, and Mahatma Gandhi National Rural Employment Guarantee Scheme, have failed to offer adequate financial assistance to support the older persons' requirements, particularly to those in unorganized working sector and below poverty line [34–36]. Rashtriya Swasthya Bima Yojana was also introduced to provide health insurance to the workforce engaged in unorganized sector but unable to perform well because it failed to capture below poverty line families, tribal blocks, and impoverished sections of the society [37–39]. To achieve the sustainable development goal of universal health coverage, the government of India has approved Ayushman Bharat Yojana (ABY) in March 2018. It is an ambitious scheme to provide financial health protection for 500 million Indian population belonging to vulnerable sections. However, like every health program in India, the success of ABY lie on overcoming existing issue like public and private sector governance, quality control, stewardship and health system organization [40].

Unfortunately, the majority of the Indian older population are unable to access healthcare services after the retirement age (60 years and above) due to paucity and poor coverage of universal health and pension programmes. Therefore, to manage the livelihood and healthcare needs, older people are compelled to work after the retirement age [19, 41]. Census of India (2011) figures illustrate that a large proportion (33 million) of older people are working after the retirement age, especially in informal sector. However, informal sector provides financial support to only marginal level of workforce after the retirement [20]. Even in formal sector, nearly 10 percent of the population engaged in selected organized work places receive the benefit of social or voluntary health insurance schemes [42]. The absence of proper financial and health care schemes can hamper the health conditions of older people indulged in economic activities. It makes older workforce as one of the most vulnerable groups in India.

Thus, given the rising magnitude of older workers, it becomes essential to understand the extent to which their health conditions are associated with the type of employment. However, no studies till date, nationally or internationally, have emphasised on this aspect. Keeping this

in mind, the present research focused on health outcomes/conditions, specific to older workers who are engaged in formal and informal sector of employment. It hypothesized that people engaged in informal employment work will experience more unfavourable health outcomes, that is, high rates of chronic health conditions, functional limitations, and poor cognitive functioning, compared to those in formal sector of employment. Research findings will help in addressing the important policy issues considering the extent of variation in health among older workers in India [41, 43].

## Research framework

Persisting health problems among older people constitutes a longstanding concern for researchers and policy makers as the prevalence of morbidity is relatively high in later ages. It is acknowledged that older people continue to participate in the labour force despite of the health risks and socio-economic challenges, particularly in developing nations [2, 19]. Kalwij, Kapteyn (2016) stated that health is multidimensional in nature and the effect of the work engagement on health varies with health indicators assessed. Evidently, prior studies have discovered that work engagement has a pronounced effect on physical [11, 12, 44–50] and mental health [10–12, 46–48, 50–52]. Few studies asserted that engagement in low paid jobs negatively influences physical health of the older people [12, 44]. Besides, working longer with degraded health conditions can have severe repercussions such as the need for long-term care, mental health issues and functional disability [11, 12]. Nevertheless, prolonged older age work engagement has also a beneficial effect on mental health [10–12, 47, 50]. This holds true in the study of Japan and Korea which describes that working in later life is generally associated with financial security and strong social network leading to better cognitive functioning and less depressive symptoms [11, 50]. However, the relationship between type of work and health could also be affected by self-selection bias in which a person may self-select into the type of work due to pre-existing health conditions, even from younger age. Meaning that there may be two-way relationship where the pre-existing health conditions influencing the type of employment and vice versa. In the case of India, the self-section into poor jobs is implausible because formal and informal sector types of work encapsulate a broad range of occupations. While individuals might select the type of occupation (within the formal or informal sector) due to their pre-existing health status, it is unlikely that they will select between formal and informal sector activities and continue to do so at an older age.

The relationship between work engagement and health conditions may vary by type of occupation [51, 53, 54]. This relationship is also shaped by various socio-economic and demographic attributes [2, 5, 12, 18–20, 44, 51, 52, 55], work characteristics [11, 12, 55–57], lifestyle behaviours [11, 16, 57–59] as well as childhood health conditions [60]. Altogether, previous studies reflected a robust relationship between work engagement and health, and taking these into account, a research framework has been conceptualized. In addition, strong relationships within health indicators also have been described in previous studies. For instance, chronic health conditions (CHC) amplifies the functional limitations [16, 49, 61, 62], and influences cognitive functioning of older people [11, 49, 51, 63]. However, the functional limitations and poor cognitive functioning are closely related to each other, as evident from previous studies which reported that physical disabilities or functional limitations increase the risk of poor cognitive functioning in older persons. Indeed, Rajan, Hebert (2013) [64], Chodosh, Miller-Martinez (2010) [65] elaborates that functional limitation plays a key in amplifying the risk of cognitive decline through neurodegenerative processes. Likewise, poor cognitive is associated with high likelihood aggregated functional limitations [66–68]. McGuire, Ford (2006) mentioned that older people with lower level of cognitive are more likely to become physically

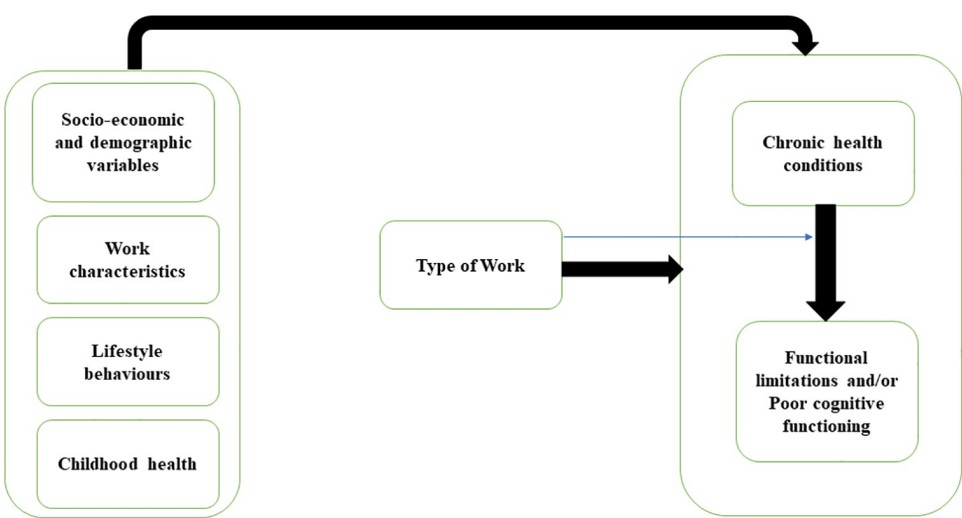

**Fig 1. Research framework.** Note: Blue line represent moderation process.

disabled than those with high cognition. Based on the findings of these studies, the combined variable of poor cognitive functioning and/or functional limitation is constructed to search for a stronger relationship between type of work, physical and cognitive functioning. As discussed above CHC may influence the functional limitations as well as poor cognitive functioning, and could modify the association between type of work, functional limitations, and poor cognitive functioning. So, the effect moderation of CHC on type of work and other health-outcomes has also been assessed in the study. The research framework reflecting the relationships between type of work and health outcomes is depicted below (Fig 1), along with adjustment factors, including childhood health, lifestyle behaviours, socio-economic and demographic characteristics as well as work characteristics.

## Materials and methods

### Study design

The present study has employed nationally representative data of Longitudinal Ageing Study in India (LASI). It is based on a cross-sectional design as it uses only one wave (baseline currently available) of the LASI data.

### Data source

The LASI baseline survey (2017–18) was conducted with the joint collaboration of International Institute for Population Sciences, Harvard TH Chan School of Public Health, and the University of Southern California [69]. This LASI survey followed a multistage stratified cluster sample design and collected information on 72,250 Indian adults aged 45 years and above, across all states and union-territories. Out of which, the sample in this study contains 31,464 older individuals (60 years and above) where around 10,746 older people (60 years and above) are currently engaged in the workforce.

The LASI survey aimed to collect the longitudinal data of older adults which include the information on social and economic wellbeing, burden of diseases, functional health and healthcare based on internationally comparable research design and tools. It created a foundation for reliable and acceptable data for national policy and long-term scientific research.

Further, it provided in-depth information of economically active older population, workforce participation across older ages and different sectors, perceived economic security, work characteristics, vulnerability, and expectations.

## Outcome variables

Following the research framework, the present study has emphasized on four health outcomes i.e., chronic health conditions (CHC), functional limitations (FL), poor cognitive functioning (PCF) and PCF and/or FL. Details of these health outcomes are given below:

**1. Chronic health conditions.** The CHC is summarized using nine self-reported health conditions. These health conditions are hypertension, diabetes, cancer, chronic lung disease, chronic heart diseases, stroke, arthritis, neurological problems, and high cholesterol. The format used in the questionnaire is "Has any health professional ever diagnosed you with any of the chronic conditions or diseases......?". Based on this information, CHC is categorized into 0 (none) and 1 (at least one health condition), where 0 labeled as 'No' and 1 as 'Yes'.

**2. Functional limitations.** The FL is constructed using 13 everyday activities which are generally termed as activities of daily living (ADL) and instrumental activities of daily living (IADL). Out of the 13 activities, the first 6 are related to ADL, while 7 are associated with IADL. The question format of FL is "Because of health or memory problem, do you have any difficulty with any of the activities...?". The FL is dichotomized as 0 (no limitations) and 1 (at least one limitation), where 0 labeled as 'No' and 1 as 'Yes'.

**3. Poor cognitive functioning.** For the assessment of poor cognitive functioning health outcome, the present research has followed the LASI report which uses the cognitive module of Health and Retirement Study (HRS) involving memory (0–20), orientation (0–8), retrieval fluency (0–61), arithmetic function (0–9), executive function (0–4), and object naming domains (0–2). First, these indicators are normalized by employing following formula:

$$\frac{(observation_i - minimum)}{(maximum - minimum)}, \quad \text{for } i = 1, 2, \ldots\ldots, n(10, 746).$$

This normalization helps in rescaling the indicators between 0 to 1 [70]. Then, a principal component analysis (PCA) is applied to create a composite score of cognitive functioning, where minimum score represents poor cognitive while higher score reflects better cognitive functioning [71, 72]. This score is further divided into three equal parts (tertile groups), where first tertile is coded as 1 (to represent poor cognitive) while rest are coded as 0 (otherwise).

**4. PCF and/or FL.** The PCF and/or FL variable is constructed by combining Poor Cognitive functioning and Functional limitation health outcomes. Below is description of PCF and/or FL:

$$PCF\_FL = \begin{cases} 1, & if \ PCF = 1 \ and/or \ FL = 1 \\ 0, & otherwise \end{cases}$$

## Type of work

LASI survey follows International Classification of Occupation 2015 to categorize the occupation types. These categories are further grouped into formal and informal work based on the guidelines provided by 66[th] round of National Sample Survey Organization report which adopts the National Classification of Occupation 2004 [73, 74]. The type of work variable is dichotomized as 0 (formal) and 1 (informal).

## Other independent variables

As per the research framework, four main dimensions of covariates are considered, that is, socio-economic and demographic, work characteristics, life-style behavior, and childhood health. Socio-economic and demographic dimension includes gender (male, female), age groups (60–65, 65+), caste groups- based on the access to wealth, power and privilege (general, Scheduled Tribe (ST), Scheduled Caste (SC), Other Backward Class (OBC)), religion (Hindu, Muslim, others), educational level (low, middle, high), marital status (currently married, others), place of residence (rural, urban), wealth (low, medium, high), and household size (1–3, 4–7, 8+). Work characteristics include working hours per week (less than 24 hours, 24–48 hours, 48+ hours), duration of being in current work (less than 15 years, 15–30 years, 30–45 years, 45 years and above) and monthly wages. The life-style behavior variables are drinking alcohol (no, yes), smoking/consuming tobacco (no, yes), physical activities consisting vigorous activities (never, rare, everyday), moderate activities (never, rare, everyday), and yoga/pranayam (never, rare, everyday). Childhood health variable includes 5 categories (very good, good, fair, poor, very poor) which have been recoded as 1. Good/fair (very good, good, fair), 0. poor (poor, very poor). Apart from these variables, the region is also taken as independent covariate which involves North (Jammu & Kashmir, Himachal Pradesh, Punjab, Uttarakhand, Haryana, Delhi, Rajasthan), Central (Uttar Pradesh, Chhattisgarh, Madhya Pradesh), East (Bihar, West Bengal, Jharkhand, Odisha), North-East (Arunachal Pradesh, Nagaland, Manipur, Mizoram, Tripura, Meghalaya, Assam), West (Gujarat, Maharashtra, Goa), South (Andhra Pradesh, Karnataka, Kerala, Tamil Nadu, Telangana), and Union territories (Chandigarh, Daman & Diu, Dadar & Nagar Haveli, Lakshadweep, Puducherry, Andaman & Nicobar).

## Statistical analysis

Descriptive summary statistics were compiled using proportions and means with associated standard deviation. The study utilized bivariate analysis with chi-square test of association and multivariable analysis following the research framework to investigate the relationships depicted in the conceptual framework (Fig 1).

First the percentage of formal and informal older workers aged 60 year and above are calculated by socio-economic and demographic variables. Then, the prevalence rate of CHC, FL, PCF and PCF and/or FL are calculated by type of work and chi-square test is applied to measure the significance of association. Lastly, sequential multiple logistic regression models are employed as the outcome variables are dichotomous. In the first model (Model-1), chronic health conditions (CHC) outcome is the function of type of work only, while in the second and third models, the effect of type of work on CHC is assessed by controlling for socio-economic and demographic variables (Model-2), and work characteristics, life-style behavior, childhood health and regions (Model-3) respectively. For functional limitation (FL) outcome, Model-1 includes the type of work and CHC residual, while in Model-2, the interaction term CHC residual*Type of work is added along with socio-economic and demographic indicators. Selection of CHC residual as a control is done by exploiting control function approach suggested by Wooldridge (2015) [75], there by instrumenting the CHC residual for model-3 in Table 1. Similarly, Model-3 adjusts for work and life-style characteristics, childhood health and regions, as in the case of CHC. For PCF, PCF and/or FL, the same procedure is followed as in case of FL. Below is the description of all models by health outcomes (Table 1), following the research framework (Fig 1).

**Table 1. Description of logistic regression models.**

| Outcome variables | Independent variables | | |
|---|---|---|---|
| | Model-1 | Model-2 | Model-3 |
| CHC | TOW | TOW, SED | TOW, SED, WC, LSB, CH, Regions |
| FL | Res_CHC, TOW | Res_CHC, TOW, Res_CHC*TOW, SED | Res_CHC, TOW, Res_CHC*TOW, SED, WC, LSB, CH, Regions |
| PCF | Res_CHC, TOW | Res_CHC, TOW, Res_CHC*TOW, SED | Res_CHC, TOW, Res_CHC*TOW, SED, WC, LSB, CH, Regions |
| PCF /and FL | Res_CHC, TOW | Res_CHC, TOW, Res_CHC*TOW, SED | Res_CHC, TOW, Res_CHC*TOW, SED, WC, LSB, CH, Regions |

Notes: CHC: chronic health conditions; FL: functional limitations; PCF: poor cognitive functioning; TOW: type of work; SED: socio-economic and demographic variables; WC: work characteristics; LSB: lifestyle behavior, CH: Childhood health; Res_CHC is obtained from logistic regression model-3 of CHC i.e., CHC–CHC_cap

## Results

### Descriptive analysis results

Table 2 shows that approximately one-third of the older population is currently working, out of which 73.1 percent are engaged in informal employment activities. Moreover, in terms of unfavorable health outcomes, the levels of CHC, FL and PCF turn out to be 43.3%, 41.6%, and 29.1% respectively. When combined, the level of PCF and/or FL amounts to 54.2% for the older workers.

Predominantly, the percentage of older workforce is highest among males, Hindus, OBC group and in rural areas. More than three-fourth of working population belongs to lower education level, while one-third of them lies in low wealth status. Around 64% of older workers have been engaged in labor force for more than 30 years. Further, only 2.6% of older workers had poor health condition during their childhood. Drinking alcohol and smoking/consuming tobacco among older workforce is 13.1 percent and 50.8 percent respectively. Geographically, older workers are more concentrated in South, East, and Central regions of India.

### Bivariate analysis results

When health outcomes were examined by type of work, as shown in Fig 2, formal older workers suffer from high burden of CHC as compared to informal counterparts (47.1% vs. 41.9%, $p < 0.0001$). On the other hand, the risks of FL, PCF, and PCF /and FL are more prevalent among informal workers (FL: 42.5% vs. 39.3%, $p < 0.05$; PCF: 32.5% vs. 20.0%, $p < 0.0001$; PCF /and FL: 57.1% vs. 46.3%, p<0.0001).

Table 3 presents the estimated rates of sampled older workforce of LASI data by the type of work they engaged in. From the table, it is observed that majority of female workforce are engaged in informal activities, conversely among male population major share lies in formal activities. In context of caste groups, the rate for informal activities is high among SC and ST, while general and OBC holds notable portion in formal activities. The percentage for informal workers is substantial among Muslim and other communities compared to Hindu community. Education and wealth play a significant role in defining nature of work. Low level of financial wellbeing and education reflects high share of work engagement in informal activities. As expected, the level of informal older workers is quite considerable in rural areas, whereas urban areas have a significant share of formal workers. Besides, concentration of informal workers is more in North-East followed by South and Western regions of India. Further, marital status and working hours reflects a meagre difference in determining type of work.

**Table 2. Background characteristics of older workers (60 years and above), N = 10,746 (34.15%).**

| Health outcomes: Type of work; Covariates | Percentage |
|---|---|
| **Health Outcomes** | |
| **Chronic Health Conditions: Yes** | 43.36 |
| **Functional Limitations: Yes** | 41.65 |
| **Poor Cognitive Functioning: Yes** | 29.13 |
| **PCF and/or FL: Yes** | 54.17 |
| **Main predictor** | |
| **Type of work: Informal** | 73.12 |
| **Socio-economic and demographic indicators** | |
| **Gender: Female** | 32.41 |
| **Age groups: 65+** | 45.99 |
| **Caste groups** | |
| General | 21.53 |
| Scheduled Tribe | 10.91 |
| Scheduled Caste | 21.17 |
| Other Backward Class | 46.39 |
| **Religion** | |
| Hindu | 82.83 |
| Muslim | 11.10 |
| Others | 6.07 |
| **Education level** | |
| Low | 78.17 |
| Middle | 16.30 |
| High | 5.52 |
| **Marital status: Others** | 25.16 |
| **Place of residence: Urban** | 21.12 |
| **Wealth** | |
| Low | 40.73 |
| Medium | 34.92 |
| High | 24.35 |
| **Household size** | |
| 1–3 | 38.14 |
| 4–7 | 47.96 |
| 8+ | 13.90 |
| **Work Characteristics** | |
| **Working hours** | |
| Less than 24 hours | 27.49 |
| 24–48 hours | 42.19 |
| 48+ hours | 30.31 |
| **Duration being in current work** | |
| Less than 15 years | 19.02 |
| 15–30 years | 16.92 |
| 30–45 years | 31.90 |
| 45 years and over | 32.16 |
| **Average wage** | 5727.70 |
| **Life-style Behaviour** | |
| **Drinking Alcohol: Yes** | 13.06 |
| **Smoking/Consuming Tobacco: Yes** | 50.87 |

(*Continued*)

**Table 2.** (Continued)

| Health outcomes: Type of work; Covariates | Percentage |
|---|---|
| **Physical Activity** | |
| **Vigorous** | |
| Never | 42.28 |
| Rare | 21.38 |
| Everyday | 36.34 |
| **Moderate** | |
| Never | 29.32 |
| Rare | 15.09 |
| Everyday | 55.59 |
| **Yoga/Pranayam** | |
| Never | 87.60 |
| Rare | 4.55 |
| Everyday | 7.85 |
| **Childhood Health: Poor** | 2.56 |
| **Regions** | |
| North | 9.14 |
| Central | 20.04 |
| East | 23.44 |
| Northeast | 2.76 |
| West | 19.09 |
| South | 25.35 |
| Union Territories | 0.17 |

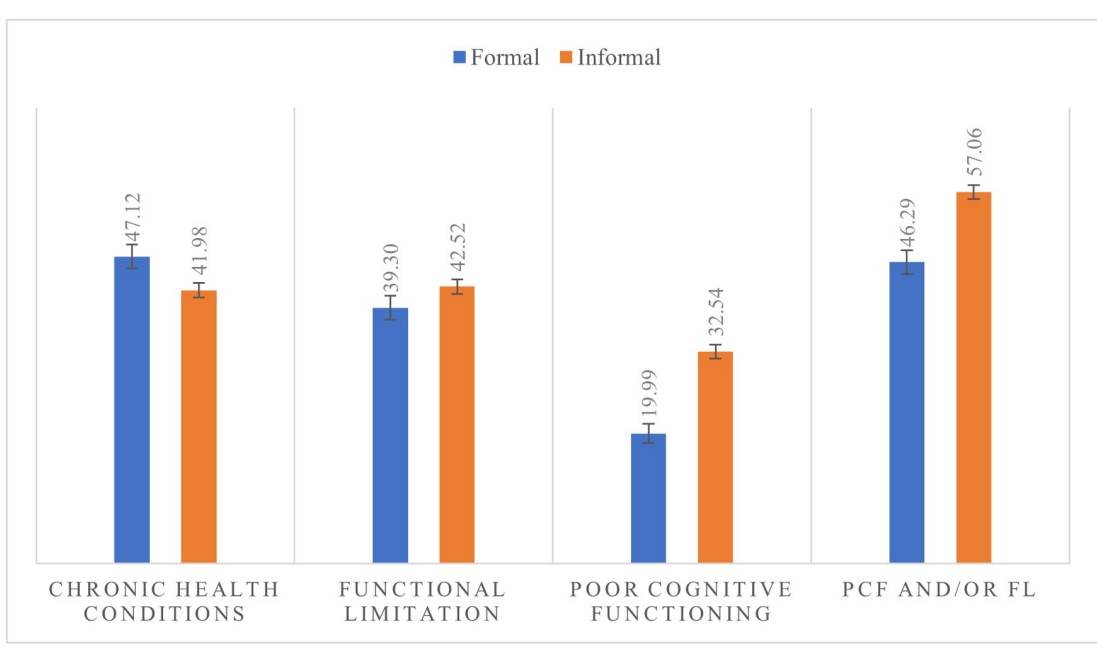

**Fig 2. Level of health outcomes by type of work.**

**Table 3. Type of work by socio-economic and demographic characteristics.**

| Covariates | Formal (%) | Informal (%) | Total working population |
|---|---|---|---|
| **Gender** | | | |
| Male | 29.67 | 70.33 | 7,311 |
| Female | 21.04 | 78.96 | 3,435 |
| **Age groups** | | | |
| 60–65 | 26.57 | 73.43 | 5,942 |
| 65+ | 27.23 | 72.77 | 4,804 |
| **Caste groups** | | | |
| General | 32.61 | 67.39 | 2,340 |
| Scheduled Tribe | 23.83 | 76.17 | 2,183 |
| Scheduled Caste | 23.45 | 76.55 | 1,922 |
| Other Backward Class | 26.50 | 73.50 | 4,301 |
| **Religion** | | | |
| Hindu | 27.76 | 72.24 | 8,079 |
| Muslim | 22.88 | 77.12 | 1,036 |
| Others | 22.04 | 77.96 | 1,631 |
| **Education level** | | | |
| Low | 22.92 | 77.08 | 8,440 |
| Middle | 36.10 | 63.90 | 1,701 |
| High | 55.68 | 44.32 | 605 |
| **Marital status** | | | |
| Currently married | 27.74 | 72.26 | 8,168 |
| Others | 24.30 | 75.70 | 2,578 |
| **Place of residence** | | | |
| Rural | 24.59 | 75.41 | 8,029 |
| Urban | 35.40 | 64.60 | 2,717 |
| **Wealth** | | | |
| Low | 22.36 | 77.64 | 4,173 |
| Medium | 26.51 | 73.49 | 3,735 |
| High | 34.96 | 65.04 | 2,838 |
| **Household size** | | | |
| 1–3 | 24.13 | 75.87 | 4,098 |
| 4–7 | 25.77 | 74.23 | 5,154 |
| 8+ | 29.79 | 70.21 | 1,494 |
| **Working hours** | | | |
| Less than 24 hours | 27.97 | 72.03 | 3,115 |
| 24–48 hours | 24.76 | 75.24 | 4,749 |
| 48+ hours | 28.83 | 71.17 | 2,882 |
| **Duration being in current work** | | | |
| Less than 15 years | 25.26 | 74.74 | 2,007 |
| 15–30 years | 27.77 | 72.23 | 1,786 |
| 30–45 years | 28.10 | 71.90 | 3,366 |
| 45 years and over | 23.60 | 76.40 | 3,394 |
| **Regions** | | | |
| North | 26.02 | 73.98 | 1443 |
| Central | 35.73 | 64.27 | 1510 |
| East | 26.75 | 73.25 | 2032 |
| Northeast | 18.03 | 81.97 | 1464 |

*(Continued)*

**Table 3.** (Continued)

| Covariates | Formal (%) | Informal (%) | Total working population |
|---|---|---|---|
| West | 25.87 | 74.13 | 1183 |
| South | 22.03 | 77.97 | 2335 |
| Union Territories | 26.45 | 73.55 | 779 |
| **Total** | **26.88** | **73.12** | **10,746** |

## Chronic health conditions

Table 4 depicts the risk of CHC among older working population in India (crude and adjusted). Informal workers tend to have less odds of CHC compared to formal counterparts. Indeed, their odds are 0.82 folds less (OR = 0.816, 95% CI: 0.748–0.890, p<0.0001). The odds ratios change slightly and remain significant after controlling socio-economic and demographic variables (in Model-2), and other covariates (in Model-3).

Additionally, among the many covariates, the model indicates that 65+ age-group is associated with increased risk of CHC. The likelihood of CHC is significantly low among older workers from ST group. On the other hand, this risk expands for Muslim and other religious communities. The odds of CHC significantly inflates with the increase in education and wealth. Similarly, urban workers are having 1.679 times (OR = 1.679, 95%CI: 1.508–1.869, p<0.0001) more odds of CHC than rural older workers. The long working hours is significantly associated with low risk of CHC. Moreover, the physical activities do have significant influence over chronic conditions but with varying directions. Vigorous activities tend to have low risk of CHC, while those who perform Yoga/Pranayam on daily basis have 1.201 times (p<0.05) more odds of chronic conditions. Similarly, poor childhood health is also significantly associated with high risk of CHC. Across India, the likelihood of CHC considerably high in South region, while it is low in Central and North-Eastern regions as compared to North.

## Functional limitations

Table 5 exhibits the relationships between FL, CHC residual and type of work, and shows that both CHC residual and type of work are significant risk factors for FL. From Model-1, the odds of FL are 1.110 times more among informal workers in contrast to formal counterparts (p<0.0001) after controlling for CHC residual, while the odds of FL increases by 1.580 times with increase in CHC residual (p<0.0001) after controlling for the type of work. Further, interacting CHC residual with type of work (Model-2 & Model-3) reveals that both type of work and CHC residual maintain their significant level. Indeed, informal workers without CHC residual (CHC residual = 0) have 0.9 times less odds of FL in contrast to formal workers without CHC, adjusted for all the covariates (OR = 0.898, 95% CI: 1.352–1.621, p<0.05). In Model-3, for formal workers, the odds of FL increases by 1.525 (p<0.0001) as one unit increase in CHC residual. Further, the odds of FL for informal workers for per unit increase in CHC residual is 1.512 (OR:1.525*0.992, 95% CI: 1.038–2.205). It shows that, effect of CHC residual is almost similar in both formal as well as informal workers since the differential effect or multiplicative factor (0.992) is close to 1 and non-significant. Meaning that the type of work does not modify the effect of CHC on FL.

Apart from these key results, among all covariates, it appeared that females are more prone to FL, while educated, wealthy and ST population are less likely to suffer from the same. Further, high odds of FL are common among those who are engaged in unhealthy lifestyle behaviours such as drinking alcohol and consuming tobacco or smoking. Besides, the odds of FL are

**Table 4. Odds of multiple logistic regression for CHC.**

| Covariates | Model-1 | Model-2 | Model-3 |
|---|---|---|---|
| **Type of work** | | | |
| Formal® | | | |
| Informal | 0.816**** (0.748 0.890) | 0.881** (0.801 0.970) | 0.891** (0.810 0.981) |
| *Socio-economic & demographic* | | | |
| **Gender** | | | |
| Male® | | | |
| Female | | 1.174*** (1.061 1.30) | 1.098 (0.980 1.231) |
| **Age groups** | | | |
| 60–65® | | | |
| 65+ | | 1.440**** (1.324 1.565) | 1.389**** (1.272 1.517) |
| **Caste groups** | | | |
| General® | | | |
| Scheduled Tribe | | 0.639**** (0.538 0.758) | 0.580**** (0.495 0.679) |
| Scheduled Caste | | 0.889 (0.775 1.019) | 0.898 (0.783 1.030) |
| Other Backward Class | | 0.975 (0.867 1.097) | 0.902 (0.806 1.010) |
| **Religion** | | | |
| Hindu® | | | |
| Muslim | | 1.221** (1.048 1.423) | 1.303**** (1.128 1.506) |
| Others | | 1.220** (1.026 1.452) | 1.235** (1.064 1.432) |
| **Education level** | | | |
| Low® | | | |
| Middle | | 1.213*** (1.075 1.369) | 1.144** (1.012 1.293) |
| High | | 1.464*** (1.209 1.774) | 1.281** (1.049 1.563) |
| **Marital status** | | | |
| Currently married® | | | |
| Others | | 1.038 (0.934 1.153) | 1.024 (0.921 1.140) |
| **Place of residence** | | | |
| Rural® | | | |
| Urban | | 1.863**** (1.671 2.077) | 1.679**** (1.508 1.869) |
| **Wealth** | | | |
| Low® | | | |
| Medium | | 1.372**** (1.239 1.519) | 1.369**** (1.235 1.517) |
| High | | 1.732**** (1.533 1.957) | 1.711**** (1.518 1.929) |
| **Household size** | | 0.939** (0.896 0.985) | 0.942** (0.899 0.987) |
| *Work Characteristics* | | | |
| **Working hours** | | | |
| Less than 24 hours® | | | |
| 24–48 hours | | | 0.919** (0.830 1.016) |
| 48+ hours | | | 0.837*** (0.747 0.938) |
| **Ln(Wage)** | | | 1.013 (0.964 1.064) |
| **Duration being in current work** | | | |
| Less than 15 years® | | | |
| 15–30 years | | | 0.972 (0.848 1.116) |
| 30–45 years | | | 0.909 (0.805 1.026) |
| 45 years and over | | | 0.895 (0.791 1.013) |
| *Life style behaviour* | | | |
| **Drinking Alcohol** | | | |

(*Continued*)

**Table 4.** (Continued)

| Covariates | Model-1 | Model-2 | Model-3 |
|---|---|---|---|
| No[®] | | | |
| Yes | | | 0.928 (0.820 1.049) |
| **Smoking/Consuming Tobacco** | | | |
| No[®] | | | |
| Yes | | | 0.996 (0.908 1.092) |
| *Physical Activity* | | | |
| **Vigorous** | | | |
| Never[®] | | | |
| Rare | | | 0.747**** (0.664 0.840) |
| Everyday | | | 0.751**** (0.677 0.833) |
| **Moderate** | | | |
| Never[®] | | | |
| Rare | | | 1.044 (0.913 1.193) |
| Everyday | | | 0.943 (0.849 1.047) |
| **Yoga/Pranayam** | | | |
| Never[®] | | | |
| Rare | | | 1.223** (1.002 1.492) |
| Everyday | | | 1.201** (1.038 1.388) |
| **Childhood health** | | | |
| Good/Fair[®] | | | |
| Poor | | | 1.694*** (1.306 2.197) |
| **Regions** | | | |
| North[®] | | | |
| Central | | | 0.530**** (0.449 0.625) |
| East | | | 0.896 (0.772 1.040) |
| Northeast | | | 0.650**** (0.540 0.782) |
| West | | | 1.162 (0.983 1.374) |
| South | | | 1.425**** (1.226 1.656) |
| Union Territories | | | 1.072 (0.880 1.306) |
| **Constant** | 0.917 | 0.767 | 0.553 |
| **AIC** | **14729.10** | **13912.61** | **13473.61** |

Note:

[®] reference category;

****(P<0.0001),

***(P<0.01),

**(P<0.05), Ln: Natural Log

greater among those who are not currently married (OR: 1.293, p<0.0001) and have poor childhood health (OR: 1.361, p<0.05). Finally, across India, the odds of FL are relatively considerable in Eastern, Western, Southern regions and in Union Territories compared to North.

## Poor cognitive functioning

From Table 6, it is observed that the type of work is significantly associated with PCF after controlling for CHC residual (Model-1). Indeed, the odds of PCF (adjusted for type of work) decreases by 0.875 (p<0.01) with one unit increase in CHC residual. Moreover, the odds of

**Table 5. Odds of multiple logistic regression for FL.**

| Covariates | Model-1 | Model-2 | Model-3 |
|---|---|---|---|
| **Res_CHC** | 1.580**** (1.454 1.717) | 1.629**** (1.365 1.944) | 1.525**** (1.280 1.817) |
| **Type of work** | | | |
| Formal[R] | | | |
| Informal | 1.110****(1.013 1.215) | 0.896** (0.810 0.992) | 0.898** (1.352 1.621) |
| **Type of work***Res_CHC | | 0.911 (0.743 1.118) | 0.992 (0.811 1.214) |
| *Socio-economic & demographic* | | | |
| **Gender** | | | |
| Male[R] | | | |
| Female | | 1.786**** (1.611 1.979) | 1.968**** (1.756 2.207) |
| **Age groups** | | | |
| 60–65 | | | |
| 65+ | | 1.468**** (1.345 1.602) | 1.421**** (1.299 1.555) |
| **Caste groups** | | | |
| General[R] | | | |
| Scheduled Tribe | | 1.177 (0.989 1.402) | 0.802*** (0.684 0.940) |
| Scheduled Caste | | 0.955 (0.826 1.105) | 0.891 (0.773 1.028) |
| Other Backward Class | | 1.035 (0.912 1.174) | 0.928 (0.823 1.046) |
| **Religion** | | | |
| Hindu[R] | | | |
| Muslim | | 1.001 (0.854 1.174) | 1.195** (1.030 1.387) |
| Others | | 0.944 (0.789 1.131) | 0.821** (0.705 0.957) |
| **Education level** | | | |
| Low[R] | | | |
| Middle | | 0.570**** (0.498 0.652) | 0.635**** (0.555 0.726) |
| High | | 0.431**** (0.342 0.543) | 0.512**** (0.406 0.647) |
| **Marital status** | | | |
| Currently married[R] | | | |
| Others | | 1.289**** (1.158 1.434) | 1.293**** (1.163 1.438) |
| **Place of residence** | | | |
| Rural[R] | | | |
| Urban | | 0.610**** (0.542 0.687) | 0.660**** (0.588 0.740) |
| **Wealth** | | | |
| Low[R] | | | |
| Medium | | 0.795**** (0.715 0.884) | 0.798**** (0.719 0.885) |
| High | | 0.820**** (0.721 0.932) | 0.832*** (0.734 0.943) |
| **Household size** | | 1.043 (0.993 1.096) | 1.059** (1.009 1.110) |
| *Work Characteristics* | | | |
| **Working hours** | | | |
| Less than 24 hours[R] | | | |
| 24–48 hours | | | 1.021 (0.922 1.132) |
| 48+ hours | | | 0.949 (0.844 1.066) |
| **Ln(Wage)** | | | 0.904*** (0.858 0.952) |
| **Duration being in current work** | | | |
| Less than 15 years[R] | | | |
| 15–30 years | | | 1.031 (0.894 1.190) |
| 30–45 years | | | 1.003 (0.884 1.137) |
| 45 years and over | | | 1.082 (0.953 1.228) |

*(Continued)*

**Table 5.** (Continued)

| Covariates | Model-1 | Model-2 | Model-3 |
|---|---|---|---|
| *Life style behaviour* | | | |
| **Drinking Alcohol** | | | |
| No[®] | | | |
| Yes | | | 1.220*** (1.078 1.380) |
| **Smoking/Consuming Tobacco** | | | |
| No[®] | | | |
| Yes | | | 1.256**** (1.142 1.382) |
| *Physical Activity* | | | |
| **Vigorous** | | | |
| Never[®] | | | |
| Rare | | | 0.957 (0.849 1.079) |
| Everyday | | | 0.861*** (0.774 0.959) |
| **Moderate** | | | |
| Never[®] | | | |
| Rare | | | 1.084 (0.944 1.246) |
| Everyday | | | 0.937 (0.839 1.046) |
| **Yoga/Pranayam** | | | |
| Never[®] | | | |
| Rare | | | 1.103 (0.896 1.357) |
| Everyday | | | 0.987 (0.845 1.152) |
| **Childhood health** | | | |
| Good/Fair[®] | | | |
| Poor | | | 1.361** (1.048 1.767) |
| *Regions* | | | |
| North[®] | | | |
| Central | | | 1.027 (0.866 1.218) |
| East | | | 1.282*** (1.096 1.500) |
| Northeast | | | 0.869 (0.713 1.057) |
| West | | | 1.833**** (1.540 2.183) |
| South | | | 1.969**** (1.682 2.305) |
| Union Territories | | | 1.263** (1.025 1.556) |
| **Constant** | 0.566 | 0.640 | 0.838 |
| **AIC** | **13737.50** | **12611.66** | **12550.70** |

Note:

[®] reference category;

****(P<0.0001),

***(P<0.01),

**(P<0.05), Ln: Natural Log

PCF (adjusted for CHC residual) are 1.7 folds among informal workers (OR = 1.683, 95%CI: 1.514–1.870, p < 0.0001; compared to formal).

In Model-3, the odds of PCF are 1.155 times for informal workers without CHC residual (CHC residual = 0) as compared formal workers without CHC residual (OR = 1.155, 95%CI: 1.152–1.637, p<0.05) after controlling for covariates. However, CHC residual loses its significance level after adding the interaction term and controlling socio-economic and demographic variables in model-2, and work-characteristics, life-style behavior, and childhood health in

**Table 6. Odds of multiple logistic regression for PCF.**

| Covariates | Model-1 | Model-2 | Model-3 |
|---|---|---|---|
| **Res_CHC** | 0.875*** (0.799 0.959) | 0.982 (0.785 1.229) | 1.000 (0.796 1.257) |
| **Type of work** | | | |
| Formal[R] | | | |
| Informal | 1.683**** (1.514 1.870) | 1.220**** (1.083 1.375) | 1.155** (1.152 1.637) |
| **Type of work*Res_CHC** | | 0.965 (0.751 1.240) | 0.922 (0.714 1.191) |
| *Socio-economic & demographic* | | | |
| **Gender** | | | |
| Male[R] | | | |
| Female | | 2.605**** (2.338 2.902) | 2.811**** (2.472 3.195) |
| **Age groups** | | | |
| 60–65 | | | |
| 65+ | | 1.478**** (1.338 1.634) | 1.370**** (1.232 1.523) |
| **Caste groups** | | | |
| General[R] | | | |
| Scheduled Tribe | | 1.783**** (1.502 2.116) | 1.813**** (1.506 2.183) |
| Scheduled Caste | | 1.128 (0.954 1.334) | 1.128 (0.948 1.341) |
| Other Backward Class | | 0.953 (0.822 1.105) | 0.939 (0.804 1.096) |
| **Religion** | | | |
| Hindu[R] | | | |
| Muslim | | 0.938 (0.784 1.123) | 1.025 (0.850 1.235) |
| Others | | (0.800** (0.688 0.930) | 0.996 (0.833 1.190) |
| **Education level** | | | |
| Low[R] | | | |
| Middle | | 0.101**** (0.074 0.139) | 0.111**** (0.081 0.153) |
| High | | 0.066**** (0.031 0.139) | 0.080**** (0.037 0.170) |
| **Marital status** | | | |
| Currently married[R] | | | |
| Others | | 1.447**** (1.289 1.624) | 1.404**** (1.248 1.581) |
| **Place of residence** | | | |
| Rural[R] | | | |
| Urban | | 0.425**** (0.371 0.487) | 0.449**** (0.388 0.520) |
| **Wealth** | | | |
| Low[R] | | | |
| Medium | | 0.549**** (0.490 0.616) | 0.551**** (0.489 0.621) |
| High | | 0.409**** (0.354 0.473) | 0.418**** (0.358 0.488) |
| **Household size** | | 1.093*** (1.037 1.153) | 1.096*** (1.038 1.158) |
| *Work Characteristics* | | | |
| **Working hours** | | | |
| Less than 24 hours[R] | | | |
| 24–48 hours | | | 0.799**** (0.709 0.900) |
| 48+ hours | | | 0.826*** (0.720 0.949) |
| **Ln(Wage)** | | | 0.824**** (0.775 0.877) |
| **Duration being in current work** | | | |
| Less than 15 years[R] | | | |
| 15–30 years | | | 1.074 (0.901 1.279) |
| 30–45 years | | | 1.061 (0.910 1.236) |
| 45 years and over | | | 1.283*** (1.106 1.489) |

*(Continued)*

**Table 6.** (Continued)

| Covariates | Model-1 | Model-2 | Model-3 |
|---|---|---|---|
| *Life style behaviour* | | | |
| **Drinking Alcohol** | | | |
| No[R] | | | |
| Yes | | | 1.510**** (1.308 1.744) |
| **Smoking/Consuming Tobacco** | | | |
| No[R] | | | |
| Yes | | | 0.957 (0.856 1.070) |
| *Physical Activity* | | | |
| **Vigorous** | | | |
| Never[R] | | | |
| Rare | | | 1.001 (0.869 1.153) |
| Everyday | | | 0.923 (0.812 1.050) |
| **Moderate** | | | |
| Never[R] | | | |
| Rare | | | 0.614**** (0.519 0.726) |
| Everyday | | | 0.629**** (0.551 0.718) |
| **Yoga/Pranayam** | | | |
| Never[R] | | | |
| Rare | | | 0.857 (0.654 1.124) |
| Everyday | | | 0.729*** (0.593 0.896) |
| **Childhood health** | | | |
| Good/Fair[R] | | | |
| Poor | | | 0.616** (0.424 0.893) |
| *Regions* | | | |
| North[R] | | | |
| Central | | | 0.835 (0.681 1.024) |
| East | | | 1.013 (0.838 1.224) |
| Northeast | | | 0.724*** (0.573 0.914) |
| West | | | 1.254** (1.016 1.548) |
| South | | | 1.072 (0.885 1.297) |
| Union Territories | | | 0.898 (0.695 1.161) |
| **Constant** | 0.252 | 0.261 | 2.217 |
| **AIC** | **12084.2** | **9878.5** | **9491.6** |

Note:

[R] reference category;

****(P<0.0001),

***(P<0.01),

**(P<0.05), Ln: Natural Log

model-3. Moreover, results show that the interaction or differential effect is non-significant, meaning that the effect of CHC residual in both formal and informal workers is quite similar and non-significant.

Among all covariates, a noteworthy gap can be seen among female workers in the odds of PCF when compared to male workers (Model-3). Likewise, the high odds of PCF are significant among ST groups, 65+ cohort and currently not married population as well as those who consume alcohol and working for more than 45 years. The odds of PCF increase as well

with the increase in household size. On the other hand, the risk decreases with rise in wealth, educational level, working hours, wage, and movement from rural to urban areas as shown by the estimated odds ratios. Nevertheless, physical activities play important role in improving cognitive function as evident from the odds of moderate exercise and Yoga/Pranayam. Geographically, the risk of PCF is high as indicated by the estimated odds ratio (OR = 1.254, p<0.05) in West region and low (OR = 0.724, p<0.01) in North-Eastern region in comparison to North.

## Poor cognitive functioning and/or functional limitations

Table 7 exhibits the relationship between type of work and PCF and/or FL. Result shows that both the type of work and CHC does significantly affect PCF and/or FL in model-1 and model-2. Informal older workers have 1.439 times (p<0.0001) more odds of PCF and/or FL compared to formal older workers. This relationship remains significant after controlling for socio-economic and demographic variables in model-2 but loses its significant level after controlling for lifestyle-behaviour, work characteristics and childhood health in model-3. In case of CHC residual, the odds of PCF and/or FL increase with increase in one unit of CHC residual in model -1. Further, after adding interaction term type of work and CHC residual, the odds of PCF and/or FL among formal older workforce increases with one unit increase in CHC residual (model-2: 1.358, p<0.01; model-3: 1.346, p<0.01). Finally, the interaction term between CHC and type of work is non-significant, and the multiplicative factor estimated through the interaction is close to 1, meaning that the effect of CHC is roughly similar in both formal and informal workers (i.e., no effect modification).

Apart from these key results, among all covariates, it appeared that females are more prone to PCF and/or FL, while educated and wealthy are less likely to suffer from the same. Further, high odds of FL are common among those who are engaged in unhealthy lifestyle behaviours such as drinking alcohol, but health lifestyle such as rigorous or moderate physical activity reduces the risk of PCF and/or FL. Besides, the odds of PCF and/or FL are 1.378 (p<0.0001) times greater among those who are not currently married. Finally, across India, the odds of PCF and/or FL are relatively considerable in Eastern, Southern and Western regions compared to North.

## Sensitivity analyses

Sensitivity analyses have also been performed by gender, place of residence and age-groups, focussing on the association between the type of work and each of the four health outcomes. Tables of results are attached as supplementary materials. The summary of the main results is given below:

1. For CHC—the effect of type of work is significant only in male (S1 Table in S1 Appendix) and 60–65 age group (S3 Table in S1 Appendix) with formal workers having higher odds of CHC.

2. For FL—the type of work, without CHC residual (CHC residual = 0), exhibit significant effects on FL (less odds of FL for informal compared to formal workers) within females (S4 Table in S1 Appendix) and 65 + age group (S6 Table in S1 Appendix) only. Moreover, the effect of CHC is much stronger among informal male workers (OR = 1.585) than female workers (OR = 1.412), and even much stronger among informal urban workers (OR = 1.857) than rural workers (OR = 1.449), while within age groups, the effect is much stronger among formal 60–65 aged workers (OR = 1.667) than 65+ aged workers (OR = 1.340) (S4 to S6 Tables in S1 Appendix).

**Table 7. Odds of multiple logistic regression for PCF and/or FL.**

| Covariates | Model-1 | Model-2 | Model-3 |
|---|---|---|---|
| **Res_CHC** | 1.249**** (1.152 1.354) | 1.358*** (1.138 1.619) | 1.346*** (1.125 1.609) |
| **Type of work** | | | |
| Formal[R] | | | |
| Informal | 1.439**** (1.318 1.572) | 1.094** (0.992 1.206) | 1.033 (0.934 1.143) |
| **Type of work*Res_CHC** | | 0.986 (0.804 1.208) | 0.970 (0.789 1.192) |
| *Socio-economic & demographic* | | | |
| **Gender** | | | |
| Male[R] | | | |
| Female | | 2.530**** (2.283 2.805) | 2.761**** (2.455 3.105) |
| **Age groups** | | | |
| 60–65 | | | |
| 65+ | | 1.530**** (1.403 1.669) | 1.435**** (1.309 1.573) |
| **Caste groups** | | | |
| General[R] | | | |
| Scheduled Tribe | | 1.165** (1.003 1.353) | 1.200** (1.020 1.410) |
| Scheduled Caste | | 0.955 (0.829 1.099) | 0.921 (0.797 1.065) |
| Other Backward Class | | 0.982 (0.875 1.103) | 0.897 (0.794 1.013) |
| **Religion** | | | |
| Hindu[R] | | | |
| Muslim | | 1.056 (0.912 1.223) | 1.154** (0.992 1.343) |
| Others | | 0.732**** (0.640 0.838) | 0.854** (0.732 0.995) |
| **Education level** | | | |
| Low[R] | | | |
| Middle | | 0.404**** (0.355 0.459) | 0.434**** (0.380 0.495) |
| High | | 0.334**** (0.267 0.418) | 0.384**** (0.304 0.484) |
| **Marital status** | | | |
| Currently married[R] | | | |
| Others | | 1.419**** (1.272 1.582) | 1.378**** (1.233 1.541) |
| **Place of residence** | | | |
| Rural[R] | | | |
| Urban | | 0.530**** (0.477 0.588) | 0.545**** (0.485 0.611) |
| **Wealth** | | | |
| Low[R] | | | |
| Medium | | 0.643**** (0.581 0.713) | 0.638**** (0.574 0.709) |
| High | | 0.564**** (0.500 0.635) | 0.586**** (0.516 0.665) |
| **Household size** | | 1.047** (0.997 1.098) | 1.077*** (1.025 1.132) |
| *Work Characteristics* | | | |
| **Working hours** | | | |
| Less than 24 hours[R] | | | |
| 24–48 hours | | | 0.904** (0.814 1.004) |
| 48+ hours | | | 0.898 (0.798 1.011) |
| **Ln(Wage)** | | | 0.864**** (0.820 0.911) |
| **Duration being in current work** | | | |
| Less than 15 years[R] | | | |
| 15–30 years | | | 0.978 (0.847 1.130) |
| 30–45 years | | | 0.916 (0.807 1.040) |
| 45 years and over | | | 1.125 (0.989 1.280) |

(*Continued*)

**Table 7.** (Continued)

| Covariates | Model-1 | Model-2 | Model-3 |
|---|---|---|---|
| *Life style behaviour* | | | |
| **Drinking Alcohol** | | | |
| No[®] | | | |
| Yes | | | 1.444**** (1.275 1.635) |
| **Smoking/Consuming Tobacco** | | | |
| No[®] | | | |
| Yes | | | 1.094 (0.994 1.204) |
| *Physical Activity* | | | |
| **Vigorous** | | | |
| Never[®] | | | |
| Rare | | | 1.000 (0.795 1.054) |
| Everyday | | | 0.865*** (0.776 0.965) |
| **Moderate** | | | |
| Never[®] | | | |
| Rare | | | 0.915 (0.795 1.054) |
| Everyday | | | 0.809**** (0.724 0.905) |
| **Yoga/Pranayam** | | | |
| Never[®] | | | |
| Rare | | | 1.021 (0.828 1.258) |
| Everyday | | | 0.952 (0.816 1.111) |
| **Childhood health** | | | |
| Good/Fair[®] | | | |
| Poor | | | 1.106 (0.843 1.451) |
| *Regions* | | | |
| North[®] | | | |
| Central | | | 0.969 (0.818 1.147) |
| East | | | 1.335**** (1.141 1.562) |
| Northeast | | | 0.878 (0.725 1.065) |
| West | | | 1.563**** (1.307 1.869) |
| South | | | 1.754**** (1.496 2.057) |
| Union Territories | | | 1.058 (0.857 1.306) |
| **Constant** | 0.801 | 0.927 | 2.992 |
| **AIC** | **14391.0** | **12691.9** | **12482.6** |

Note:

[®] reference category;

****(P<0.0001),

***(P<0.01),

**(P<0.05), Ln: Natural Log

3. For PCF—informal workers exhibit higher odds of PCF within both male and female, rural and urban areas, and age groups. However, the effects of type of work are significant only within male (S7 Table in S1 Appendix), rural area (S8 Table in S1 Appendix) and 65+ age group (S9 Table in S1 Appendix) workers.

4. For PCF and/or FL—The effect of CHC is significant within male and female, although slightly stronger among informal male workers (OR = 1.370) than formal male workers

(OR = 1.334) (S10 Table in S1 Appendix). Moreover, the effect of CHC is also significant within 60–65 age group only, but much stronger among formal workers (OR = 1.452) than informal workers (OR = 1.365) (S12 Table in S1 Appendix). Finally, the effect of CHC is also significant within both rural and urban residence areas, but much stronger among urban workers (informal: OR = 1.653; formal: OR = 1.449) (S11 Table in S1 Appendix).

## Discussion

In the backdrop of increasing older population in India and the paucity of pension/ financial benefits schemes, it is quite likely that this population is still working after the retirement age. The present study supports this argument and observed that around one-third of the population aged 60 years and above are still in labour force. However, majority of this population are engaged in informal activities contributing nearly 73 percent of total older population.

Besides, the share of labour force participation is higher among males than that of females. On the other hand, the participation rate of female population in informal activities is greater compared to male counterparts. Certainly, females are generally expected to spend most of the time in household activities and taking care of their family members. Consequently, they choose low paid elementary activities which needs less work time [2, 20, 51, 52]. Likewise, the engagement in informal work is highest among Muslim community, Scheduled Caste, and Scheduled Tribe groups. It is well documented that these sections of the society in India are marginalized due to their poor socio-economic conditions. Reddy (2016) explains that socio-economically disadvantage populations tend to have high labour-force participation rate in later ages. Further, pronounced level of participation in informal sector can be observed among older population with low education and wealth. This large workforce is mostly engaged in agricultural activities, casual labour or unskilled occupations which require less education [2, 20]. Generally, agricultural activities in India are concentrated in rural areas and involve subsistence farming, resulting in high proportion of workforce participation in later life.

Previous studies demonstrated that work engagement has a pronounced effect on physical and mental health. Consistent with these articles, the present research reaffirms that work engagement does play a significant role in determining unfavourable health outcomes. Nevertheless, current findings also provide the evidence of varied unfavourable health outcomes by type of economic activity which has not been documented in former research. In fact, informal older workers are less likely to suffer from CHC and FL, while more likely to have PCF as compared to formal older workers. The emergence of high CHC rates is most likely to be observed among educated and wealthy groups [22, 25, 76] which is closely attributed to socio-economic profiles of formal workers. Nonetheless, noticeable change in nature of relationship is observed between type of work and FL. In model-1, the risk of FL is substantially high among informal workers, but after controlling for the covariates in model-3, the risk of FL changed drastically reflecting favourable effect of informal activities on FL for older people with no chronic health conditions. The low risk of FL among informal workers confirms healthy work effect concept suggested by former literatures [48, 52, 57, 77] because informal activities involve physically demanding works and only those can be engaged in later life who are not suffering from severe physical functioning.

Continuous work engagement in later life may have leading implication on cognitive functioning [11, 48, 50, 51]. Extending from this relationship, the present research also illustrates varied influence of type of work on PCF. For instance, informal workers without any CHC are more likely to suffer from PCF as compared to formal workers without CHC. Likewise, the combined health conditions such as PCF and/or FL are relatively high among informal workers after adjusting for CHC. However, this relationship is not significant after controlling for

work-characteristics, lifestyle behaviour and childhood health conditions. The major factor which distorts the relationship between type of work and PCF and/or FL is unhealthy lifestyle behaviour such as drinking alcohol. Further, the risk of PCF and/or FL increases among formal workers with increase in CHC and remains consistent after controlling for all the covariates. From this finding, it can be inferred that unhealthy lifestyle may lead to serious health implication in later life of older workers especially among formal sector workers with CHC.

From the sensitivity analysis results, it can be observed that the risk of CHC is significantly low among informal male and 60–65 age group workers. Moreover, FL is less prevalent among informal female and 65+ age group workers. Conversely, the risk of PCF is relatively high among informal male, rural areas, and 65+ age group working population. The reason of low functional limitations among informal females and 65+ age group workers compared to their formal counterparts is an example of "healthy work effect" [48, 52, 57, 77], which means female and 65+ age group informal workers are already in better physical condition to continue their economic activity.

Further, this study has certain limitations such as use of cross-sectional data at a single point of time. Thus, result of the study could only provide the evidence of statistical association between type of work and health outcomes not the cause-effect relationship. To get the better picture of type of work and health, longitudinal data would be a better option. Thus, future studies based on cause-effect relationship between type of work and health outcomes would provide a platform for preventive strategies to deal with health-related issues of working older population.

## Conclusion

Older population in India constitutes an undeniable share of labour-force after the retirement age of 60 years. This extent is the outcome of financial constraints caused by paucity in social and health insurance schemes. Moreover, working longer impacts physical and mental health of the older people which varies by formal or informal sector of employment. Further, improving health conditions of this vulnerable population should be an utmost priority for policy makers to encourage active and healthy ageing.

Therefore, the present study underscores the relevance of policies focusing on providing health and healthcare benefits by respective economic activity and socio-economic position. Further, policies should also put emphasis on promotion of healthy lifestyle behaviour among older workers. Adequate working conditions should be considered during policy formulation, which can offer a level of job satisfaction and contribute to better well-being. Moreover, the preference of employment type should be given precedence according to the age-groups. It would be better that older adults aged 70+ should participate in part-time jobs as they may be unsuitable to handle high strain and physically demanding work. Economic security policy should be recommended to those who are lacking physical capacity. This can help them to sustain their livelihood without any financial constraints. Additionally, campaigns should be promoted by the government and other social bodies to create awareness regarding benefits physical activity and drawbacks of consuming alcohol on mental and physical health in later life. The issues of older workers in India should be taken seriously otherwise it will lead to a huge chunk of vulnerable groups with inadequate social and financial support.

## Supporting information

**S1 Appendix.**
(DOCX)

## Acknowledgments

I humbly thank International Institute for Population Sciences (IIPS) for giving me access to Longitudinal Ageing Study of India dataset for this research.

### Ethical approval

This research was performed in compliance with all applicable laws and institutional guidelines. Ethical approval was obtained from the University of Canberra's Human Research Ethics Committee (reference number: 202211511).

## Author Contributions

**Conceptualization:** Poulomi Chowdhury, Itismita Mohanty, Theo Niyonsenga.

**Data curation:** Poulomi Chowdhury.

**Formal analysis:** Poulomi Chowdhury.

**Methodology:** Poulomi Chowdhury, Itismita Mohanty, Theo Niyonsenga.

**Resources:** Theo Niyonsenga.

**Supervision:** Itismita Mohanty, Theo Niyonsenga.

**Validation:** Itismita Mohanty, Akansha Singh, Theo Niyonsenga.

**Writing – original draft:** Poulomi Chowdhury.

**Writing – review & editing:** Itismita Mohanty, Akansha Singh, Theo Niyonsenga.

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
