## [Decision Letter · Decision Letter 0]

15 Jun 2022

PONE-D-22-08253Informal Sector Employment and the Health Outcomes of Older Workers in IndiaPLOS ONE

Dear Dr. Chowdhury,

Thank you for submitting your manuscript to PLOS ONE. After careful consideration, we feel that it has merit but does not fully meet PLOS ONE’s publication criteria as it currently stands. Therefore, we invite you to submit a revised version of the manuscript that addresses the points raised during the review process.

The referees and I see value in this paper, given the novelty that it brings to the literature on how working after retirement can affect individual health, which, and I do agree with the referees on this point, is a very interesting and important topic.  Given the clear expertise of the referees, I defer to their comments and will ask you to simply respond to them. However, my view is that the main areas of concerns, which you would do well to address in rewriting the paper, are as follows:

I agree with the reviewers that the main drawback of the current version of the paper it is that your empirical strategy does not take into account self-selection of individuals. Even though you are looking for associations and not to causal relationships - and this should be made clear in the paper and the language used should be adapted accordingly- I believe that the issues raised by the referees, i.e. self-selection and reverse causality- should be discussed, and documented as much as possible together with potential biases suffered by your coefficients, as suggested by reviewer 2. In particular, I agree with reviewer 1 that control functions could provide a viable and helpful solutions to control for self-selection and should be used in the baseline estimates. These estimates could be added as a robustness exercise should it turn out that the estimates do not change when controlling for self-selection. For more details, please refer to the comments of the Reviewers reported below.I believe that controlling for initial conditions in health, as suggested by reviewer 1, will provide another useful robustness exercise. You should discuss the problems your identification strategy suffers due to the lack of longitudinal data, and provide evidence of how robust are your results to the inclusion of initial conditions.Finally, separate estimates by gender will effectively acknowledge differences existing in the selection process related to gender and clustering will provide more reliable standard errors, as suggested by reviewer 2.

Beyond the above highlighted points, as I noted previously the referee reports are all of high quality, so please make sure to respond directly to all the comments.

Congratulations on the work so far; I look forward to reading the revision.

We look forward to receiving your revised manuscript.

Kind regards,

Simona Lorena Comi

Academic Editor

PLOS ONE

Journal Requirements:

Reviewers' comments:

Reviewer's Responses to Questions

**Comments to the Author**

1. Is the manuscript technically sound, and do the data support the conclusions?

Reviewer #1: Yes

Reviewer #2: Yes

2. Has the statistical analysis been performed appropriately and rigorously? 

Reviewer #1: Yes

Reviewer #2: Yes

3. Have the authors made all data underlying the findings in their manuscript fully available?

Reviewer #1: No

Reviewer #2: No

4. Is the manuscript presented in an intelligible fashion and written in standard English?

Reviewer #1: Yes

Reviewer #2: Yes

5. Review Comments to the Author

Reviewer #1: Review of the paper “Informal Sector Employment and the Health Outcomes of Older Workers in India.”

General Comment

The paper focuses on an important research question: how does working after retirement age in the formal and informal sector expose older people to health events. The topic is crucial in the context of India, where a large proportion of the population is not covered by social security and bears on a family network to live at older ages.

The paper takes advantage of the LASI survey and evaluates the association between working in the formal and informal sector after age 60 and health outcomes. The main results suggest that individuals in the informal sectors have higher chances of poor cognitive functioning, whereas working in the formal sector is associated with more chronic health conditions and functional limitations. The study calls for policy able to protect older workers by providing health healthcare benefits.

I think this paper brings a good perspective on an important policy question and takes advantage of both survey data and econometric techniques to address this question. However, the fact that these results are only an association between the field of work and health events, thus causality cannot be claimed, makes the findings relatively weak and could not provide sufficient ground for a robust empirical analysis. I leave to the editor the decision whether the paper, after major revisions, is sufficiently original to be considered for publication.

Major Points to be addressed.

As a general comment, I think the authors should address the role of selection in their analysis: those who work over the retirement age are usually either at the top or bottom of the income distribution for different reasons. In particular, those who are at the bottom of the income distribution indeed are those who need to work to make their ends meet. In this context, if the sample includes primarily individuals working in the informal/formal sector belonging to the bottom of the income distribution, I suspect the role of selection: since health status is correlated with socioeconomic conditions, you are looking at people with lower health capital and attributing to them an effect of working longer on their health status, which is instead due to an initial lower health endowment. Thus, you are looking at selected individuals who are likely to have poor health status in general.

Another important question which has not been addressed is the reverse association (given that we cannot claim causality) between the type of work and health conditions. What if individuals with cognitive deficits, limitations in functioning (and chronic diseases) are more likely to find a job in the informal (or formal) sector and work after age 60 (to make their ends meet)? Thus, if this is the case, then it could be that having pre-existing health conditions predicts the chance of working in certain types of jobs, which is associated with higher chances of having these health conditions lately.

Overall, I think that the paper should clarify from the beginning that this study looks at the association between formal/informal sectors and health outcomes, and it is not claiming causality.

Pg 5 of the manuscript: “However, no study till date, nationally or internationally, has emphasised this aspect (concerning the extent to which the health conditions are associated with the type of employment for older workers). I argue that the following reference looks at this type of association, which is not cited in the paper.

Nag, A., Vyas, H., & Nag, P. (2016). Occupational health scenario of Indian informal sector. Industrial health, 2015-0112.

Although they do not specifically focus on the older workers, their sample includes a vast population of workers. Thus, I think that this study should be acknowledged at least.

Page 6: in Figure 1, I think another critical piece of information to be taken into account is the initial health endowment: following the seminal contribution of Grossman (1972), health is part of the human capital of each individual; thus, the initial health endowment should be taken into account when evaluating the effect of the type of work on late-life health effects.

For example, the authors should consider including health as a child. To avoid the omitted variable bias problem, I advise the author to include the information about the childhood health conditions available in the LASI questionnaire on questions HT231, HT232, HT233 HT234. This information is much needed to be considered to control for the initial health endowment.

Page 7: There is a reference to the total sample of the survey, but how many individuals are aged above 60? The text-only reports those who are still in the workforce.

Section 2. (page 8) Functional Limitations: there is no reference to which ones are the 13 specific activities

Type of work section (page 9): there is no reference to the complete list of classification of occupation nor in the appendix. Al least an example should be provided to allow the reader to get a grasp of the classification.

Methods (page 7 onwards): Although this study looks at the association between type of work and health outcomes using logistic regression models, I wonder why the authors did not provide alternative models such as probabilistic or ols models. In this context, I also argue that the choice of measuring CHC as a function of the polynomial in Model-3 would suggest the need to instrument this variable. Thus, when looking at FL and PCF outcomes, including the CHC as control could be done by exploiting the Control Function approach à la Wooldridge (2015), thus instrumenting the CHC with residuals of Model-3 in Table-1.

Finally, are the results reported in Tables 4 to 6 coefficients or marginal effects? To compare results across Tables, marginal effects should be displayed to have a sense of the magnitude.

Table-2: The sample composition is not gender-balanced. This might have affected the results. I also wonder whether the place of residence, which is proportioned toward the rural area, has again some drawbacks such as self-selection of those individuals who are poorer and less educated (in line with the 78% of individuals in the sample reporting low education level).

The role of household size appears to be particularly important in Indian society. I would suggest the author show the distribution of household size greater than 4, given that it includes 61.34% of the sample.

Bivariate analysis results (page 13): I suggest adding the confidence intervals to Figure 2.

Tables from 4 (page 15): as a general comment, I wonder why doing yoga could increase the chances of CHC and how the authors explain this finding.

As a general comment, I would suggest the authors conduct sensitivity analyses by subgroups of individuals, namely by gender, geographical area, and age group 60-64 and 65+. This could confirm the robustness of the results.

Another alternative classification related to formal vs informal jobs is the separation between non-strenuous vs strenuous jobs. This could be investigated as an alternative.

Minor Points to be addressed

Page 3: When reporting “India, …., reportedly 8.6 percent (104 million) of the total population in 2011…” I think a reference should be added.

Page 6: Why there is the name of the authors reported next to reference number (7)? I think that the manuscript should show consistency with the style of referencing.

References

Nag, A., Vyas, H., & Nag, P. (2016). Occupational health scenario of Indian informal sector. Industrial health, 2015-0112.

Wooldridge (2015), Control function methods in applied econometrics, Journal of Human Resources, 50(2), 420-445.

Reviewer #2: This paper uses the baseline wave of longitudinal Ageing Study in India (LASI) to investigate the association between health and employment in later years, as well as the heterogeneous effect by employment type. The analytical sample contains only working populations over age 60. Overall, the paper is well written in a clear and scientific matter. Empirical results consistently show the health effect varies by employment type and specific health outcomes. Employment type significantly moderates the effect of CHC on FL, and FL on PCF.

Comments:

My major comments are on the complexity of employment in later years as established by existing research (e.g., Rietveld et al.,2014; van Zon et al.,2020). It is understandable that the cross-sectional nature of the dataset limits further exploration of any dynamic aspects of employment, also, the goal of the paper is not aimed at a causal identification. However, it could be more informative to add some relevant descriptive statistics, sensitivity checks, or comments regarding the defined employment status that might not truly reflect the general associations.

1) The observed employment type might contain some noise due to potential complex work transitions in later years such as partial retirement. Some transitions and jobs could be quite recent and temporary, especially in the informal sector, which may require additional control for working characteristics such as working years and sectors (public/private).

Without any further details, it is possible that some people are working as a recovery from an early retirement or mandatory retirement, or at a phase of bridge job. For one thing, the current employment can differ much from their main career which determinants one’s pension and other welfare schemes.

For another, the health effect of employment might be biased given potentially complex sorting into employment for older workers. For instance, if one transited from a full-time office-based job to self-employment to better manage working hours, the estimated health effect of the informal sector might vary by time. Socioeconomically advantaged individuals might extend working life as a way to continue social participation which probably exerts a positive impact on mental health and well-being, in the meanwhile, affecting physical health.

2) It would be better to add separate estimation by gender considering the gender difference in terms of both health, healthcare utilisation and employment aspects (e.g., Kandrack et al.,1991). Especially, labour force participation is more selective among females than males, and there are also great heterogeneities across occupations within formal or informal sectors. For instance, continued employment may buffer against risk factors that aggravate women’s cognitive health (Oi, Katsuya 2019).

For other minor comments:

3) It might be better to cluster standard error at a household level to account for the potential correlation among employment/retirement decisions within a household. Old couples might have a joint retirement (or working) plan.

4) The implication about the healthcare services needs could probably be better supported by adding some descriptive statistics about healthcare utilisation, and healthcare entitlements (if data permits).

5) Potential measurement error due to cognitive impairment. To what extent the estimated might be affected by the recall error in covariates such as wealth and job characteristics? Any potential screening criteria for cognitive impairment? Are surveyed people all self-interviewed or include proxy answers?

Typos:

P17: informal workers without CHC have 0.8 folds less odds

P26: those determinants which have considerable impacts

Reference:

1) Kandrack, Mary-Anne, Karen R. Grant, and Alexander Segall. "Gender differences in health related behaviour: some unanswered questions." Social science & medicine 32.5 (1991): 579-590.

2) Oi, Katsuya. "Does gender differentiate the effects of retirement on cognitive health?." Research on Aging 41.6 (2019): 575-601.

3) Rietveld, Cornelius A., Hans van Kippersluis, and A. Roy Thurik. "Self‐employment and health: Barriers or benefits?." Health economics 24.10 (2015): 1302-1313.

4) van Zon, Sander KR, et al. "Multimorbidity and the transition out of full-time paid employment: a longitudinal analysis of the health and retirement study." The Journals of Gerontology: Series B 75.3 (2020): 705-715.

6. PLOS authors have the option to publish the peer review history of their article (what does this mean?). If published, this will include your full peer review and any attached files.

Reviewer #1: No

Reviewer #2: No

---

## [Author Response · Author response to Decision Letter 0]

30 Aug 2022

We would like to thank the reviewers for their useful comments. Please consider our responses below. Please note that all references to line numbers in this response letter relate to line numbers in the document with track changes. 

Reviewer reports:

Reviewer 1:

Major Points to be addressed:

Reviewer 1 - Comment#1: As a general comment, I think the authors should address the role of selection in their analysis: those who work over the retirement age are usually either at the top or bottom of the income distribution for different reasons. In particular, those who are at the bottom of the income distribution indeed are those who need to work to make their ends meet. In this context, if the sample includes primarily individuals working in the informal/formal sector belonging to the bottom of the income distribution, I suspect the role of selection: since health status is correlated with socioeconomic conditions, you are looking at people with lower health capital and attributing to them an effect of working longer on their health status, which is instead due to an initial lower health endowment. Thus, you are looking at selected individuals who are likely to have poor health status in general.

Another important question which has not been addressed is the reverse association (given that we cannot claim causality) between the type of work and health conditions. What if individuals with cognitive deficits, limitations in functioning (and chronic diseases) are more likely to find a job in the informal (or formal) sector and work after age 60 (to make their ends meet)? Thus, if this is the case, then it could be that having pre-existing health conditions predicts the chance of working in certain types of jobs, which is associated with higher chances of having these health conditions lately.

Overall, I think that the paper should clarify from the beginning that this study looks at the association between formal/informal sectors and health outcomes, and it is not claiming causality.

Reply: We thank the reviewer for this useful comment. 

We agree with the reviewer on the comments regarding the potential issues of selection and reverse-causation. We believe that addressing these issues will improve the overall presentation of this manuscript. First, we acknowledge the reviewer’s suggestion to clarify from the beginning that this study looks at the association between formal/informal sectors and health outcomes (page 5, lines 12-19). Based on reviewer’s comments, we also have done some modifications in our analysis (page 10, lines 12, 22-24; page 11, lines 21-25).

Studies elsewhere found that prolonged working beyond retirement age influences health conditions of the older population. In that context little has been explored on working life and its influence on the health outcomes of Indians in older age. In this study we analysed how type of formal/informal sector employment among older people (60 years and more) influences their health outcomes. While in India older population suffers from poor health conditions irrespective of their socio-economic status, few studies have focused on the sector of employment in this age group and how it influences their health. We agree with the reviewer’s concern on individuals self-selecting themselves into the formal/informal nature of employment based on their pre-existing health conditions. Meaning that there may be two-way relationship where the pre-existing health conditions influencing the type of employment and vice versa. However, this research is about how the prolonged working of older people beyond their retirement age in the formal/informal sector of employment influences their health outcomes. We believe that it is unlikely for individuals to self-selecting into the type of employment due to their pre-existing health status. We agree with the reviewer’s observation that older adults from low socio-economic status continue working if their financial circumstances do not permit them to retire. Those who are higher up in the socio-economic status and do not want to give up earning or those who are passionate about their work may continue working as well. We believe that, while individuals may potentially self-select into informal/formal sector of employment in their young working age, they do not continue to do so at an older age. Although the study sample is skewed towards low education level (78%), it does not include primarily the bottom of the income distribution (low level of wealth) which represents 40.7%, compared to 34.9% and 24.4% in medium and high wealth levels, respectively. Also, the sample information on how long people have been working in their present occupation reveals that, on average, they have been in their present occupation for 36 years, rather than choosing the sector of employment and/or shifting jobs at an older age. Moreover, formal and informal sector work types in India encapsulate a broad range of occupations and, while individuals might select the type of occupation (within the formal or informal sector) due to their pre-existing health status, it is unlikely that they will select between formal and informal sector activities. Having said that, we do not deny the fact that some people with greater physical and cognitive impairment may self-select into one of these sectors from the very beginning of their work life and not at an older age. We assume that would be a minority in our sample. Therefore, we believe that working beyond retirement age due to financial limitations and the nature of work (formal/informal) could influence older people’s health conditions which is the focus of this research. 

We are aware of the potential bias due reverse association, as we are limited to use cross-sectional data in the absence of longitudinal data. However, we believe that the use of covariates’ adjustment in the multivariable regression models performed will alleviate both self-selection and reverse association induced bias. Additionally, we have addressed and discussed these issues in the study limitation section (pages 35, lines 15 to 19). 

Reviewer 1 - Comment#2: Pg 5 of the manuscript: “However, no study till date, nationally or internationally, has emphasised this aspect (concerning the extent to which the health conditions are associated with the type of employment for older workers). I argue that the following reference looks at this type of association, which is not cited in the paper.

Nag, A., Vyas, H., & Nag, P. (2016). Occupational health scenario of Indian informal sector. Industrial health, 2015-0112.

Although they do not specifically focus on the older workers, their sample includes a vast population of workers. Thus, I think that this study should be acknowledged at least.

Reply: Thank you for the suggestion.

The paper by Nag et al. (2016) has discussed about occupational hazards and injuries of selected cases of the informal sector. Based on their finding, they have advocated for protective measurement to safeguard the workers from the harsh work environment. This paper is not directly related with our study because we are focusing on long term health conditions. However, we have cited this paper in our discussion section as some of its finding can provide support to our arguments (page 33, line 15; page 34, line 1). 

Reviewer 1- Comment#3: Page 6: in Figure 1, I think another critical piece of information to be taken into account is the initial health endowment: following the seminal contribution of Grossman (1972), health is part of the human capital of each individual; thus, the initial health endowment should be taken into account when evaluating the effect of the type of work on late-life health effects.

For example, the authors should consider including health as a child. To avoid the omitted variable bias problem, I advise the author to include the information about the childhood health conditions available in the LASI questionnaire on questions HT231, HT232, HT233 HT234. This information is much needed to be considered to control for the initial health endowment.

Reply: Thank you for the suggestion.

To avoid the omitted variable bias problem, we have incorporated childhood health conditions and the duration of current work in the study as covariates to control for. This is seen in the abstract (page 2, line 9), materials and methods (page 10, lines 12, 22-24) and in Tables 4-7.

Reviewer 1 - Comment#4: Page 7: There is a reference to the total sample of the survey, but how many individuals are aged above 60? The text-only reports those who are still in the workforce.

Section 2. (page 8) Functional Limitations: there is no reference to which ones are the 13 specific activities

Type of work section (page 9): there is no reference to the complete list of classification of occupation nor in the appendix. Al least an example should be provided to allow the reader to get a grasp of the classification.

Reply: Thank you for the comment.

We have added number of older individuals 60 years and above in data source section (page 8, line 7). We have added tables describing functional limitations (table S14) and type of work (table S13) in the appendix section.

Reviewer 1 - Comment#5: Methods (page 7 onwards): Although this study looks at the association between type of work and health outcomes using logistic regression models, I wonder why the authors did not provide alternative models such as probabilistic or ols models. In this context, I also argue that the choice of measuring CHC as a function of the polynomial in Model-3 would suggest the need to instrument this variable. Thus, when looking at FL and PCF outcomes, including the CHC as control could be done by exploiting the Control Function approach à la Wooldridge (2015), thus instrumenting the CHC with residuals of Model-3 in Table-1.

Finally, are the results reported in Tables 4 to 6 coefficients or marginal effects? To compare results across Tables, marginal effects should be displayed to have a sense of the magnitude.

Reply: Thank you for the comments and suggestions.

 The reviewer is correct. Alternative models such as probabilistic (probit/tobit) or OLS models can also be used for the analysis. However, in our analysis we wanted to compare those workers who had poor health outcome by their type of economic activity. For example, for cognitive functioning we have compared only those who have poor cognitive level to understand if person is likely to experience PCF, how the risk of PCF varies by type of work. For this purpose, we have constructed our variable in binary form (presence vs. absence) and used logistic regression model.

 We thank the reviewer for the suggestion regarding CHC residual, it really helped in improving the results of our analysis. We have incorporated CHC residual as a control variable in all our models for Functional Limitations and Poor Cognitive Functioning, instead of using the actual values for CHC (as shown in Table 5, 6 and 7, pages 20-26). The reviewer’s comment also prompted us to consider a new outcome variable which combines both poor cognitive functioning and functional limitations (PCF and/or FL) (page 8, line 19 and page 9, lines 22-24). The new variable is motivated in the research framework section (page 6, lines 23-25, page 7, lines 1-8). The following text has been added into the manuscript. 

Page 6, lines 23-25 - However, the functional limitations and poor cognitive functioning are closely related to each other, as evident from previous studies that physical disabilities or functional limitations increases the risk of poor cognitive functioning in older persons.

Page 7, lines 1-8 - Rajan, Hebert (2013), Chodosh, Miller‐Martinez (2010) elaborates that functional limitation plays a key in amplifying the risk of cognitive decline through neurodegenerative processes. Likewise, poor cognitive is associated with high likelihood aggregated functional limitations (66-68). McGuire, Ford (2006) mentioned that older people with lower level of cognitive are more likely to become physically disabled than those with high cognition. Based on the findings of these studies, the combine variable of poor cognitive functioning and/or functional limitation is constructed to provide a better relationship between type of work, physical and cognitive functioning. 

Page 9, lines 22-24 – The PCF and/or FL variable is constructed by combining Poor Cognitive functioning and Functional limitation health outcomes. Below is description of PCF and/or FL:

PCF_FL= {█(1, if PCF=1 and/or FL=1@0, otherwise)┤

Results of PCF and/or FL is given below (page 26, lines 2-15; page 27, lines 1-8):

Table-7 exhibits the relationship between type of work and PCF and/or FL. Result shows that both the type of work and CHC does significantly affect PCF and/or FL in model-1 and model-2. Informal older workers have 1.439 times (p<0.0001) more odds of PCF and/or FL compared to formal older workers. This relationship remains significant after controlling for socio-economic and demographic variables in model-2 but loses its significant level after controlling for lifestyle-behaviour, work characteristics and childhood health in model-3. In case of CHC residual, the odds of PCF and/or FL increase with increase in one unit of CHC residual in model -1. Further, after adding interaction term type of work and CHC residual, the odds of PCF and/or FL among formal older workforce increases with one unit increase in CHC residual (model-2: 1.358, p<0.01; model-3: 1.346, p<0.01). Finally, the interaction term between CHC and type of work is non-significant, and the multiplicative factor estimated through the interaction is close to 1, meaning that the effect of CHC is roughly similar in both formal and informal workers (i.e., no effect modification).

Apart from these key results, among all covariates, it appeared that females are more prone to PCF and/or FL, while educated and wealthy are less likely to suffer from the same. Further, high odds of FL are common among those who are engaged in unhealthy lifestyle behaviours such as drinking alcohol, but health lifestyle such as rigorous or moderate physical activity reduces the risk of PCF and/or FL. Besides, the odds of PCF and/or FL are 1.378 (p<0.0001) times greater among those who are not currently married. Finally, across India, the odds of PCF and/or FL are relatively considerable in Eastern, Southern and Western regions compared to North.

 Thank you for suggesting marginal effects to display the magnitude of effect of each co-variable, other covariates being held constant. But as far as we believe, odds ratio can also be used to present magnitude of effects of the covariates, like marginal effects do. With the logistic regression approach, we are modelling the risk or probability of the event (e.g., risk or probability of experiencing PCF). The choice of odds ratios is automatic as we are comparing the odds of PCF in formal and informal groups. Regression coefficients are at the log-scale; by taking their exponentiation, we reverse back to the odds scale.

Reviewer 1 - Comment#6: Table-2: The sample composition is not gender-balanced. This might have affected the results. I also wonder whether the place of residence, which is proportioned toward the rural area, has again some drawbacks such as self-selection of those individuals who are poorer and less educated (in line with the 78% of individuals in the sample reporting low education level).

The role of household size appears to be particularly important in Indian society. I would suggest the author show the distribution of household size greater than 4, given that it includes 61.34% of the sample.

Reply: Thank you for the comment.

 We have cross-checked our result through sensitivity analysis by gender, place of residence and age groups. A section on sensitivity analysis is added in the manuscript (page 29, lines 3-23; page 30, lines 1-3).

 The figure below shows the distribution of household size greater than 4. The highest of percentage of household size is concentrated between 4 to 8 members of the household. However, we re categorized the household size variable into three categories namely 1. 1-3 members, 2. 4-7 members, 3. 8+ members.

Reviewer 1 - Comment#7: Bivariate analysis results (page 13): I suggest adding the confidence intervals to Figure 2.

Reply: We thank the reviewer for this useful comment. The confidence intervals have now been added in the figure 2.

Reviewer 1 - Comment#8: Tables from 4 (page 15): as a general comment, I wonder why doing yoga could increase the chances of CHC and how the authors explain this finding.

Reply: Thank you for the comment.

Thank you for pointing out this result. The increase in CHC using Yoga is an interesting result because we are using cross-sectional data and can only justify association. So, it could be possible that, those people who are performing yoga, are more likely to be suffering from any chronic health conditions.

Reviewer 1 - Comment#9: As a general comment, I would suggest the authors conduct sensitivity analyses by subgroups of individuals, namely by gender, geographical area, and age group 60-64 and 65+. This could confirm the robustness of the results.

Another alternative classification related to formal vs informal jobs is the separation between non-strenuous vs strenuous jobs. This could be investigated as an alternative.

Reply: We thank the reviewer for this insightful comment. We believe that addressing this comment in the revised draft will improve the clarity of the results.

As mentioned in the previous comment (comment #6), the sensitivity analyses have been performed by gender, geographical area, and age groups.

The sensitivity analysis has confirmed the robustness of our result, which is described below (page29-30):

 For CHC - the effect of type of work is significant only in male (Table S1) and 60-65 age group (Table S3) with formal workers having higher odds of CHC.

 For FL - the type of work, without CHC residual (CHC residual=0), exhibit significant effects on FL (less odds of FL for informal compared to formal workers) within females (Table S4) and 65 + age group (Table S6) only. Moreover, the effect of CHC is much stronger among informal male workers (OR=1.585) than female workers (OR=1.412), and even much stronger among informal urban workers (OR=1.857) than rural workers (OR=1.449), while within age groups, the effect is much stronger among formal 60-65 aged workers (OR=1.667) than 65+ aged workers (OR=1.340) (Tables S4 to S6). 

 For PCF - informal workers exhibit higher odds of PCF within both male and female, rural and urban areas, and age groups. However, the effects of type of work are significant only within male (Table S7), rural area (Table S8) and 65+ age group (Table S9) workers.

 For PCF and/or FL - The effect of CHC is significant within male and female, although slightly stronger among informal male workers (OR=1.370) than formal male workers (OR=1.334) (Table S10). Moreover, the effect of CHC is also significant within 60-65 age group only, but much stronger among formal workers (OR=1.452) than informal workers (OR=1.365) (Table S12). Finally, the effect of CHC is also significant within both rural and urban residence areas, but much stronger among urban workers (informal: OR=1.653; formal: OR=1.449) (Tables S11).

Thank you for suggesting an alternative classification for the type of work, which we believe would provide a different insight on the association between the type of work and health outcome. This suggestion can be examined for our future research because analysis based on non-strenuous vs strenuous job classification in the present paper would deviate from our study objective which focuses on the classification based on two major economic activities. 

Minor Points to be addressed:

Reviewer 1 - Comment#10: Page 3: When reporting “India, …., reportedly 8.6 percent (104 million) of the total population in 2011…” I think a reference should be added.

Page 6: Why there is the name of the authors reported next to reference number (7)? I think that the manuscript should show consistency with the style of referencing.

Reply: Thank you for the comment.

 Reference has been added in this statement.

 All the referencing errors have been corrected.

Reviewer 2:

Reviewer 2 - Comment#1: My major comments are on the complexity of employment in later years as established by existing research (e.g., Rietveld et al.,2014; van Zon et al.,2020). It is understandable that the cross-sectional nature of the dataset limits further exploration of any dynamic aspects of employment, also, the goal of the paper is not aimed at a causal identification. However, it could be more informative to add some relevant descriptive statistics, sensitivity checks, or comments regarding the defined employment status that might not truly reflect the general associations.

1) The observed employment type might contain some noise due to potential complex work transitions in later years such as partial retirement. Some transitions and jobs could be quite recent and temporary, especially in the informal sector, which may require additional control for working characteristics such as working years and sectors (public/private).

Without any further details, it is possible that some people are working as a recovery from an early retirement or mandatory retirement, or at a phase of bridge job. For one thing, the current employment can differ much from their main career which determinants one’s pension and other welfare schemes.

For another, the health effect of employment might be biased given potentially complex sorting into employment for older workers. For instance, if one transited from a full-time office-based job to self-employment to better manage working hours, the estimated health effect of the informal sector might vary by time. Socioeconomically advantaged individuals might extend working life as a way to continue social participation which probably exerts a positive impact on mental health and well-being, in the meanwhile, affecting physical health.

Reply: We thank the reviewer for this insightful comment.

 We agree with reviewer that past work characteristics and job transition are important indicators that could have been useful in our analysis, however LASI data does not provide information pertaining to past jobs. Further, as per the Indian context, the phenomena of job transition are quite unlikely, especially a shift from formal to informal sector are not common. As per our analysis, 64% of the sample participants spent 30 years or more in their current job activity, and on average, people spent 36 years in their present job. For older population (age of 60 years and above), these statistics indicate low level of job transition. Moreover, in LASI data, public/private sector related information is only applicable for few selected formal sectors (service or salaried person).

 Without any further details, it is possible that some people are working as a recovery from an early retirement or mandatory retirement, or at a phase of bridge job. For one thing, the current employment can differ much from their main career which determinants one’s pension and other welfare schemes. We agree with the reviewer that the current employment can differ much from the main career. This might be the case for the 19% of sample participants whose duration in the current work is less than 15 years. However, in the Indian context, it is less likely that these people would be working as a recovery from retirement or a phase of bridge job. We have now cross-checked our results through sensitivity analysis by gender, age-group, and place of residence. We believe this added value to the overall presentation of the manuscript and improved clarity of our findings. We have now added this information in the appendix section from table S1-S12. From the sensitivity analysis we have found that the risk of CHC is significantly low among informal male and 60-65 age group workers. Moreover, FL is less prevalent among informal female and 65+ age group workers. Conversely, the risk of PCF is relatively high among informal male, rural areas, and 65+ age group working population.

Reviewer 2 - Comment#2: It would be better to add separate estimation by gender considering the gender difference in terms of both health, healthcare utilisation and employment aspects (e.g., Kandrack et al.,1991). Especially, labour force participation is more selective among females than males, and there are also great heterogeneities across occupations within formal or informal sectors. For instance, continued employment may buffer against risk factors that aggravate women’s cognitive health (Oi, Katsuya 2019).

Reply: Thank you for the comment.

We thank the reviewer and agree that gender specific analysis will add value to this research. We have now conducted separate gender specific analysis. Please refer to our response to your Comment#1 and Reviewer-1’s comment#9. While the objective of this research is to study the association between type of work and health outcomes. We will study, healthcare utilization by type of work in the next research objective (as part of PhD thesis) which will be covered in a different paper.

For other minor comments:

Reviewer 2 - Comment#3: It might be better to cluster standard error at a household level to account for the potential correlation among employment/retirement decisions within a household. Old couples might have a joint retirement (or working) plan.

Reply: Thank you for the comment. While we agree with the reviewer’s concern about cluster standard error at household level as it might be affecting the consistency of the coefficient estimates. However, we believe that might not be a major issue in this study as around 80% (from LASI data) of the respondents in our sample are single working members from a household. 

Reviewer 2 - Comment#4: The implication about the healthcare services needs could probably be better supported by adding some descriptive statistics about healthcare utilisation, and healthcare entitlements (if data permits).

Reply: Thank you for the comment.

We agree with the reviewer’s concerns on health service utilisation, and we are addressing that in another study as part of the PhD thesis.

Reviewer 2 - Comment#5: Potential measurement error due to cognitive impairment. To what extent the estimated might be affected by the recall error in covariates such as wealth and job characteristics? Any potential screening criteria for cognitive impairment? Are surveyed people all self-interviewed or include proxy answers?

Reply: Thank you for the comment.

In LASI, the Health and Retirement study (HRS) module has been used to derive cognitive measures. These measures are collected using various domains- including memory, orientation, retrieval fluency (verbal fluency), arithmetic, executive functioning, and object naming. To measure the cognitive functioning, the respondents were asked certain questions pertaining to the cognitive measure domains and the responses were noted by interviewer based on the feedback or reply from the respondents, so no proxy person was involved during the interview. Further, as per the LASI report, those participants were excluded from the measurement who sought or receive assistance during interview.

Reviewer 2 - - Comment#6: Typos:

P17: informal workers without CHC have 0.8 folds less odds

P26: those determinants which have considerable impacts

Reply: Thank you for the comment.

All the typo errors have been rectified in the manuscript.

---

## [Decision Letter · Decision Letter 1]

11 Oct 2022

PONE-D-22-08253R1Informal Sector Employment and the Health Outcomes of Older Workers in IndiaPLOS ONE

Dear Dr. Chowdhury,

Thank you for submitting your manuscript to PLOS ONE. After careful consideration, we feel that it has merit but does not fully meet PLOS ONE’s publication criteria as it currently stands. Therefore, we invite you to submit a revised version of the manuscript that addresses the points raised during the review process.

Thank you for submitting the revised version of your manuscript to PLOS ONE. Both reviewers and I feel that the paper is much improved. However, there are a few issues raised by the reviewers that you should address. I would like you to revise your paper, taking into account the comments of the referees (see below).

Specifically, I do agree with Reviewer 1 that self-selection into job could be an issue and should be mentioned and discussed in the paper. Furthermore, the Discussion section is not very effective and should be shortened and rewritten to better highlight your key results.

We look forward to receiving your revised manuscript.

Kind regards,

Simona Lorena Comi

Academic Editor

PLOS ONE

Journal Requirements:

Reviewers' comments:

Reviewer's Responses to Questions

**Comments to the Author**

1. If the authors have adequately addressed your comments raised in a previous round of review and you feel that this manuscript is now acceptable for publication, you may indicate that here to bypass the “Comments to the Author” section, enter your conflict of interest statement in the “Confidential to Editor” section, and submit your "Accept" recommendation.

Reviewer #1: All comments have been addressed

Reviewer #2: All comments have been addressed

2. Is the manuscript technically sound, and do the data support the conclusions?

Reviewer #1: Yes

Reviewer #2: Yes

3. Has the statistical analysis been performed appropriately and rigorously? 

Reviewer #1: Yes

Reviewer #2: Yes

4. Have the authors made all data underlying the findings in their manuscript fully available?

Reviewer #1: No

Reviewer #2: No

5. Is the manuscript presented in an intelligible fashion and written in standard English?

Reviewer #1: Yes

Reviewer #2: Yes

6. Review Comments to the Author

Reviewer #1: I think the authors have responded satisfactorily to most of the comments from the first review round. I do however still have some small comments for some of them.

In particular, I think the role of individuals with poor health self-selecting into poorer jobs, which leads to a stronger deterioration of health should be acknowledged better than as it is now in the manuscript. Although it is unlikely that individuals self-selected themselves into the type of employment due to pre-existing health status later in life, it could be that they self-select at the beginning of their working careers and they remain in poor jobs.

It is fundamental to stress that individuals could ex ante engage in poor jobs due to their precarious health status at a young age and thus leading to late-life health decline. I think this point could be better explained in the introduction, as they explain in the report which I received.

Finally, regarding the Discussion session, I found the section very dense and I would suggest stressing the key findings of the paper as well as the take-home message, in order to provide a more synthetic interpretation.

Minor Comments

Table 2 : I wonder whether for binary covariates it would be sufficient to report only one category (for example only those who have chronic health conditions, instead of both no/yes), to make the tables more compact.

Reviewer #2: Many thanks for your efforts in the revision, and I think all my concerns have been responded or addressed properly.

For the new version, I might have two additional comments. The first is the control function approach which has changed many models. If I understand correctly, it is used as a control for partial chronic conditions that are unexplained by observables. If much of variation in chronic conditions have been explained in a separate model, the residual variable might not be a good proxy for chronic conditions.

Second, I am not sure whether I grasp the idea of combining PCF and FL as the fourth outcome variable to explore the potential interactions between PCF and FL. Indeed, in the final results, the coefficients of informal worker on PCF and FL have opposite signs, making the coefficient on PCF/PF insignificant. I am not sure about the idea here or the relevant explanation.

7. PLOS authors have the option to publish the peer review history of their article (what does this mean?). If published, this will include your full peer review and any attached files.

Reviewer #1: No

Reviewer #2: No

---

## [Author Response · Author response to Decision Letter 1]

15 Nov 2022

We would like to thank the reviewers for their useful comments. Please consider our responses below. Please note that all references to line numbers in this response letter relate to line numbers in the document with track changes. 

Reviewer reports:

Reviewer 1- Comment#1: I think the authors have responded satisfactorily to most of the comments from the first review round. I do however still have some small comments for some of them. In particular, I think the role of individuals with poor health self-selecting into poorer jobs, which leads to a stronger deterioration of health should be acknowledged better than as it is now in the manuscript. Although it is unlikely that individuals self-selected themselves into the type of employment due to pre-existing health status later in life, it could be that they self-select at the beginning of their working careers and they remain in poor jobs. It is fundamental to stress that individuals could ex ante engage in poor jobs due to their precarious health status at a young age and thus leading to late-life health decline. I think this point could be better explained in the introduction, as they explain in the report which I received.

Reply: We thank the reviewer for this useful comment.

As per the reviewer’s suggestion, we have added some lines on “self-section into job” in the research framework section of manuscript (page no. 6 and line no. 13-21).

The paragraph is as follows:

“However, the relationship between type of work and health could also be affected by self-selection bias in which a person may self-select into the type of work due to pre-existing health conditions, even from younger age. Meaning that there may be two-way relationship where the pre-existing health conditions influencing the type of employment and vice versa. In the case of India, the self-section into poor jobs is implausible because formal and informal sector types of work encapsulate a broad range of occupations. While individuals might select the type of occupation (within the formal or informal sector) due to their pre-existing health status, it is unlikely that they will select between formal and informal sector activities and continue to do so at an older age.” 

Reviewer 1- Comment#2: Finally, regarding the Discussion session, I found the section very dense and I would suggest stressing the key findings of the paper as well as the take-home message, in order to provide a more synthetic interpretation.

Reply: Thank you for pointing this out, we have improved our discussion section by focusing only on the key findings of the paper, and we have also removed some paragraphs which were not that important. Further, discussion section is segregated in 5 aspects: 1. basic scenario of older workers in India, 2. association between type of work and health outcomes, 3. role of other covariates, 4. sensitivity analysis, and 5. study limitations.

Following paragraphs, we have removed from the discussion section:

1. From page no. 29, line 10-17

Consequently, agrarian households tend to have larger family size which is assumed to provide better manpower to support farming activities. This can be clearly seen through current findings showing higher work participation for larger family size households. However, it is also important to note that the level of engagement in informal activities is considerable for those households with small family sizes. Tafuro (2020) elucidated that intergenerational support is strongly related with son’s preference in countries like, India, China, Vietnam, and South Korea, and it is quite common that older people without sons are more likely to be economically active in these regions.

2. Page 31 (line 24-25) and page 32 (line 1-2)

It is because Japan is a developed country where provision for social and financial security is strong, therefore older workers only engage to maintain their social network and get the sense of pride of being economically independent. 

3. Page no. 32, line 13-19

Heterogeneity in context of health outcomes across regions can be clearly observed in India. Notably, chances of CHC upsurges in southern region, but FL is relatively intensifying in southern and western region. Geographically, there is a stark difference in terms of PCF, it rises in western region, then dwindles in north-eastern region. Further, the combine conditions of PCF and/or FL are high in eastern, western, and southern regions of India. Despite of physical and mental health limitations, older population in India are still working after the retirement age especially in informal sector.

Reviewer 1- minor comment#1: Table 2: I wonder whether for binary covariates it would be sufficient to report only one category (for example only those who have chronic health conditions, instead of both no/yes), to make the tables more compact.

Reply: Thank you for the suggestion.

Now, we have reported only the percentage of health outcomes variables, and other binary covariates in Table-2 (page no.13). 

Reviewer 2- Comment#1: Many thanks for your efforts in the revision, and I think all my concerns have been responded or addressed properly.

For the new version, I might have two additional comments. The first is the control function approach which has changed many models. If I understand correctly, it is used as a control for partial chronic conditions that are unexplained by observables. If much of variation in chronic conditions have been explained in a separate model, the residual variable might not be a good proxy for chronic conditions.

Reply: Yes, CHC residual is the unexplained partial chronic conditions which we have calculated through the following procedure:

1. We have run the full model for CHC incorporating all the covariates (Table-4, model-3).

2. Then, the predicted value of CHC is calculated.

3. Further, the residual value is calculated by subtracting predicted CHC from the observed CHC.

So, the reviewer’s understanding is correct, if most of the variation is explained in chronic conditions, taking residual variable of chronic condition should not be a good proxy. However, in our case, the CHC residual in Table-5, Table-6, and Table-7 for model-1 is coming significant, which means that there are still some unexplained variations in CHC which is also influencing our outcome variables. Therefore, we could say that taking CHC residual as a proxy for CHC is a better option. Moreover, the use of the control function was suggested by the first reviewer in the first review round. Indeed, the reviewer #1 stated: “Thus, when looking at FL and PCF outcomes, including the CHC as control could be done by exploiting the Control Function approach à la Wooldridge (2015), thus instrumenting the CHC with residuals of Model-3 in Table-1”.

Reviewer 2- Comment#2: Second, I am not sure whether I grasp the idea of combining PCF and FL as the fourth outcome variable to explore the potential interactions between PCF and FL. Indeed, in the final results, the coefficients of informal worker on PCF and FL have opposite signs, making the coefficient on PCF/PF insignificant. I am not sure about the idea here or the relevant explanation.

Reply: Thank you for the comment.

The rationale behind combining PCF and/or FL is constructed to search for a better and stronger relationship between type of work, physical and cognitive functioning. Further, previous literatures have emphasised that PCF and FL are the two-sides of the same coin because functional limitation plays a key in amplifying the risk of cognitive decline (Miller Martinez et al., 2010; Rajan et al., 2013), whereas poor cognitive is associated with high likelihood of functional limitations (McConnell et al., 2002; Dodge et al., 2005; McGuire et al., 2006). 

The coefficients of informal worker on FL, PCF, and PCF/FL (main effects, Model 1 in Tables 5, 6 & 7) have all positive signs and are statistically significant. The opposite signs for coefficients of informal worker on PCF and FL in the final model results should not be interpreted as the main effects since these models include the interaction term type of work * residual CHC. These coefficients represent the effect of type of work in the absence of any residual CHC (i.e., residual CHC = 0). Moreover, the final models have been adjusted for various covariates, some being enough important to explain some of the variation in FL, PCF, and PCF/FL. The insignificant result of type of work on PCF and/or FL may not necessarily due to the opposite signs but to other covariates controlled for in the model. It may suggest as well that the type of work could not have any influence if a person suffers from both the conditions. Further, other unhealthy life-style behaviours may play an important role in distorting the relationship between type of work and PCF and/or FL. The rationale of PCF and/or FL and reason for insignificant relationship between type of work and PCF and/or FL is already mentioned in research framework section (page no. 7 and line no. 5-15) and discussion section (page no. 30 and line no. 14-17).

Research framework (page no. 7 and line no. 5-15):

However, the functional limitations and poor cognitive functioning are closely related to each other, as evident from previous studies which reported that physical disabilities or functional limitations increase the risk of poor cognitive functioning in older persons. Indeed, Rajan, Hebert (2013), Chodosh, Miller‐Martinez (2010) elaborates that functional limitation plays a key in amplifying the risk of cognitive decline through neurodegenerative processes. Likewise, poor cognitive is associated with high likelihood aggregated functional limitations [66-68]. McGuire, Ford (2006) mentioned that older people with lower level of cognitive are more likely to become physically disabled than those with high cognition. Based on the findings of these studies, the combined variable of poor cognitive functioning and/or functional limitation is constructed to search for a stronger relationship between type of work, physical and cognitive functioning.

Discussion (page no. 30 and line no. 14-17):

However, this relationship is not significant after controlling for work-characteristics, lifestyle behaviour and childhood health conditions. The major factor which distorts the relationship between type of work and PCF and/or FL is unhealthy lifestyle behaviour namely such as drinking alcohol.

---

## [Decision Letter · Decision Letter 2]

26 Dec 2022

PONE-D-22-08253R2Informal Sector Employment and the Health Outcomes of Older Workers in IndiaPLOS ONE

Dear Dr. Chowdhury,

Thank you for submitting your manuscript to PLOS ONE. After careful consideration, we feel that it has merit but does not fully meet PLOS ONE’s publication criteria as it currently stands. Therefore, we invite you to submit a revised version of the manuscript that addresses the points raised during the review process.

My apologies for the time it has taken me to get back to you despite a very timely response from the two original reviewers. Both reviewers are overall quite happy with the progress made but reviewer 1 still wants to see a few additional changes to the discussion section before recommending an unconditional acceptance. I agree with that assessment – you have responded very well to most of the issues raised and as a result, the manuscript has taken a big step toward a final publication. However, I would still ask you to address the remaining comments of reviewer 1 in a final revision of your paper.

We look forward to receiving your revised manuscript.

Kind regards,

Simona Lorena Comi

Academic Editor

PLOS ONE

Journal Requirements:

Reviewers' comments:

Reviewer's Responses to Questions

**Comments to the Author**

1. If the authors have adequately addressed your comments raised in a previous round of review and you feel that this manuscript is now acceptable for publication, you may indicate that here to bypass the “Comments to the Author” section, enter your conflict of interest statement in the “Confidential to Editor” section, and submit your "Accept" recommendation.

Reviewer #1: All comments have been addressed

Reviewer #2: All comments have been addressed

2. Is the manuscript technically sound, and do the data support the conclusions?

Reviewer #1: Yes

Reviewer #2: Yes

3. Has the statistical analysis been performed appropriately and rigorously? 

Reviewer #1: Yes

Reviewer #2: Yes

4. Have the authors made all data underlying the findings in their manuscript fully available?

Reviewer #1: No

Reviewer #2: No

5. Is the manuscript presented in an intelligible fashion and written in standard English?

Reviewer #1: Yes

Reviewer #2: Yes

6. Review Comments to the Author

Reviewer #1: The authors have addressed all my comments. However, I want to stress that the discussion needs some adjustments. The discussion should focus on the main findings and the paper's strength and limitations, as well as potential future research. As it is now the discussion includes also the basic scenario (which is well explained in the introduction) and the role of covariates (which I do not find essential in this section). I think the authors could improve the discussion. After that, the paper would likely be ready to be accepted for publication.

Reviewer #2: Many thanks for the updated version. All my comments have been explained in general. My small concern will still be around the justification to combine CPF and FL although they are closely and positively correlated. For me, the separate results for each as outcome variable seem to be already informative about the effect of informal sector on health outcomes. The mechanisms might be different for cognition and functional limitation.

7. PLOS authors have the option to publish the peer review history of their article (what does this mean?). If published, this will include your full peer review and any attached files.

Reviewer #1: No

Reviewer #2: No

---

## [Author Response · Author response to Decision Letter 2]

23 Jan 2023

We would like to thank the reviewers for their useful comments. Please consider our responses below.

Reviewer reports:

Reviewer 1:

The authors have addressed all my comments. However, I want to stress that the discussion needs some adjustments. The discussion should focus on the main findings and the paper's strength and limitations, as well as potential future research. As it is now the discussion includes also the basic scenario (which is well explained in the introduction) and the role of covariates (which I do not find essential in this section). I think the authors could improve the discussion. After that, the paper would likely be ready to be accepted for publication.

Reply: We thank the reviewer for this useful comment. As per the suggestion of the reviewer, we have removed the points pertaining to roles of covariates from the discussion as the interpretation of other covariates is already given in result section. Following are the points which we have removed from page no. 30 (lines 8-25) to 31 (lines 1-21). 

“This research also sheds light on those determinants of unfavourable health outcomes which are socio-economic and demographic attributes, work characteristics, lifestyle behaviour, childhood health and regions of India. Among them, prolonged work engagement has detrimental effect on physical and cognitive functioning for those who are female older workers if likened to male counterparts. Commonly mainstream female workers are employed in informal activities, and they alone work as domestic workers giving rise to double burden of activities [51, 52, 78]. Additionally, it is known that women live longer than men and continue to work in their later life with one or more disabilities [11, 52, 79]. Increasing age has adverse implications on health conditions of older people [22, 80, 81], which can be observed through the result that 65+ working population tend to suffer more from physical and mental health conditions. In Indian society, ST and Muslim community are also the part of vulnerable and marginalized sections who tend to work longer in later ages. The risk of PCF is substantial among ST groups, while the burden of CHC is more prominent among Muslim community. This is may be because ST group is mostly engaged in manual labour or scavenging [82], and agricultural labour activities [83] which is associated with decline in cognitive ability [51]. On the other hand, Muslim community in India are well known for their engagement in informal artisanal work such as weaving, carpentry, black-smithing, and Zari work which is related ergonomic condition leading to CHC [78, 84-87]. However, the combination of PCF and/or FL is remarkably high among both ST group and Muslim community which reflects having these poor conditions altogether elevates the health risk for any marginalized community as compared to their well-off counterparts.

Education and wealth are considerable predictors of unfavourable health outcomes. Increase in education and wealth leads to reduction in FL, PCF, and PCF and/or FL, though escalating the chances of CHC. As evident from studies that better education and wealth improves the physical functioning and work ability [44, 45]. In case of work characteristics, increase in working hours and wages is associated with improvement in PCF, but this result is contradictory to the studies conducted in Japan which states that part time or low working hours improves the cognitive functioning of older workers [10, 11]. Although as it is cross-sectional study, the results could also reflect that only those older people are working longer who have better cognitive functioning. The life-style behaviours are essential components which needs to be taken into account to justify the relationship between type of work and health. Drinking alcohol and consuming tobacco/ smoking can lead to serious health implication, as evident from the result that it degrades physical as well as cognitive functioning. Subsequently, vigorous physical exercise diminishes the risk of CHC and FL, however Yoga/ Pranayam ameliorates cognitive functioning. Moreover, adverse impacts of CHC and FL can be minimized by staying active and maintaining a healthy lifestyle [88-90]. On the other hand, poor childhood health condition worsens the risk of CHC, and FL. Similar findings can be seen in the findings of Pavela and Latham (2016), White, Campo (2013) that poor childhood conditions are strongly associated with development of chronic conditions in later ages.”

Moreover, we have added few lines related to potential future research in page no.32 and line no. 8 to 11.

“Thus, future studies based on cause-effect relationship between type of work and health outcomes would provide a platform for preventive strategies to deal with health-related issues of working older population.”

Reviewer 2: 

Many thanks for the updated version. All my comments have been explained in general. My small concern will still be around the justification to combine PCF and FL although they are closely and positively correlated. For me, the separate results for each as outcome variable seem to be already informative about the effect of informal sector on health outcomes. The mechanisms might be different for cognition and functional limitation.

Reply: Thank you for pointing out your concern pertaining to combined variable of PCF and FL. Our rationale of PCF and/or FL is based on the previous literatures which have emphasised that PCF and FL are the two-sides of a same coin as both are related to each other. Further, we were also interested to see the influence of type of work if a person is having both the health conditions. Indeed, separate analysis have provided the informative findings, but we have also found that type of work does not play a significant when a person suffers from both the conditions. In other words, having PCF and FL together could diminishes the influence of type of work.

---

## [Decision Letter · Decision Letter 3]

7 Feb 2023

Informal Sector Employment and the Health Outcomes of Older Workers in India

PONE-D-22-08253R3

Dear Dr. Chowdhury,

We’re pleased to inform you that your manuscript has been judged scientifically suitable for publication and will be formally accepted for publication once it meets all outstanding technical requirements.

Kind regards,

Simona Lorena Comi

Academic Editor

PLOS ONE

Additional Editor Comments (optional):

Reviewers' comments:

Reviewer's Responses to Questions

**Comments to the Author**

1. If the authors have adequately addressed your comments raised in a previous round of review and you feel that this manuscript is now acceptable for publication, you may indicate that here to bypass the “Comments to the Author” section, enter your conflict of interest statement in the “Confidential to Editor” section, and submit your "Accept" recommendation.

Reviewer #1: All comments have been addressed

Reviewer #2: All comments have been addressed

2. Is the manuscript technically sound, and do the data support the conclusions?

Reviewer #1: Yes

Reviewer #2: Yes

3. Has the statistical analysis been performed appropriately and rigorously? 

Reviewer #1: Yes

Reviewer #2: Yes

4. Have the authors made all data underlying the findings in their manuscript fully available?

Reviewer #1: No

Reviewer #2: No

5. Is the manuscript presented in an intelligible fashion and written in standard English?

Reviewer #1: Yes

Reviewer #2: Yes

6. Review Comments to the Author

Reviewer #1: The authors have addressed my concerns regarding the discussion part and I think that the manuscript is now in good shape.

Reviewer #2: Many thanks for further explanation. According to the coefficients from previous Tables, the correlation between working sector and FL and PCF work in an opposite direction - reducing the significance level and the size of the coefficient in Table 7. The findings from separate regressions do suggest more interesting and probably complex relationship between PCF and FL, as well as working sector. As the results do not suggest causal effect, it seems that people working in an informal sector, like agriculture, are likely to have greater physical capacity, which in the meanwhile offsets some cognition declining. The results in this part might need more explanations. The results seem to suggest weaker relevance to the point as suggested by the literature, like any potential amplification of one problem to the other. Since this does not affect the main conclusion and results in general, I am fine with the current version

7. PLOS authors have the option to publish the peer review history of their article (what does this mean?). If published, this will include your full peer review and any attached files.

Reviewer #1: No

Reviewer #2: No

---

## [Editor Report · Acceptance letter]

10 Feb 2023

PONE-D-22-08253R3 

Informal Sector Employment and the Health Outcomes of Older Workers in India 

Dear Dr. Chowdhury:

I'm pleased to inform you that your manuscript has been deemed suitable for publication in PLOS ONE. Congratulations! Your manuscript is now with our production department. 

Kind regards, 

on behalf of

Professor Simona Lorena Comi 

Academic Editor

PLOS ONE